# Observational evidence of accelerating electron holes and their effects on passing ions

Yue Dong [1], Zhigang Yuan [1] ✉, Shiyong Huang[1], Zuxiang Xue[1], Xiongdong Yu[1], C. J. Pollock[2], R. B. Torbert [3] & J. L. Burch [4]

As a universal structure in space plasma, electron holes represent an obvious signature of nonlinear process. Although the theory has a 60-year history, whether electron hole can finally accelerate ambient electrons (or ions) is quite controversial. Previous theory for one-dimensional holes predicts that net velocity change of passing electrons (or ions) occurs only if the holes have non-zero acceleration. However, the prediction has not yet been demonstrated in observations. Here, we report four electron holes whose acceleration/deceleration is obtained by fitting the spatial separations and detection time delays between different Magnetospheric Multiscale spacecraft. We find that electron hole acceleration/deceleration is related to the ion velocity distribution gradient at the hole's velocity. We observe net velocity changes of ions passing through the accelerating/decelerating holes, in accordance with theoretical predictions. Therefore, we show that electron holes with non-zero acceleration can cause the velocity of passing ions to increase in the acceleration direction.

Electron holes observed by satellites in collisionless space plasmas are Debye-scale solitary potential structures of Bernstein–Green–Kruskal mode[1–5] and are manifestations of strongly nonlinear processes. These electrostatic solitary structures are ubiquitous in space plasmas, including the plasma sheet[6–9], bow shock[10–14], auroral region[15,16] and especially the reconnection sites[17–22]. In addition, they can exist in laboratory plasmas[22–25]. Such nonlinear structures have been thoroughly studied due to their potential for particle acceleration[17,26–29], anomalous plasma resistivity[22,30,31], and dissipating magnetic energy during reconnection[21–23,32], which are important issues in plasma physics. Previous studies have suggested electron acceleration around electron holes[17,28,29,33], but there have been no direct observational evidence of electron or ion acceleration by symmetric potential electron holes for a long time. Recently, with high-resolution observations of the Magnetospheric Multiscale (MMS) mission[34], Fu et al. found that there is no electron acceleration and no obvious electron distribution changes around a single electron hole[35].

Previous studies and discussions are based on electron holes drifting at a constant speed, but according to theoretical research,

electron holes with acceleration can change the net momentum of electrons or ions passing through them[36]. Some electron holes are observed to have drift speeds comparable to local ion thermal speed[6,7,17,18,35,37,38], so they are called "slow electron holes". Ambiguously, earlier theory and simulations indicated that slow electron holes cannot exist stably because the interaction with ambient ions will prevent their velocities from remaining at or below the typical ion thermal speed[39–42]. However, recent statistical observations revealed that slow electron holes with speeds near the local minimum of a double-humped background ion velocity distribution function (VDF) can avoid acceleration caused by the interaction with ions and thus exist stably[38]. Although theory predicts that slow electron holes may accelerate or decelerate depending on kinetic features of the ion VDF[43], the acceleration of electron holes has not been demonstrated in observations. In this letter, we search for accelerating (or decelerating) slow electron holes and investigate their effect on the ion VDF with the help of the high-resolution observations of the MMS spacecraft.

In this work, we present MMS observations of accelerating electron holes and demonstrate that electron holes with non-zero

[1]School of Electronic Information, Wuhan University, Wuhan, China. [2]Denali Scientific, Fairbanks, AK, USA. [3]Physics Department, University of New Hampshire, Durham, NH, USA. [4]Southwest Research Institute, San Antonio, TX, USA. ✉e-mail: y_zgang@vip.163.com

acceleration can cause a net velocity change of ions passing through them. Combining the observations and theoretical analysis on the accelerating/decelerating slow electron holes, we propose a schematic illustration of the interaction between the slow electron hole and the local ion VDF (Fig. 1). First, the velocity of the electron hole ($v_{EH}$) is affected by the local ion VDF: when $v_{EH}$ is at the positive (negative) gradient of the ion VDF, the electron holes are decelerated (accelerated). This result naturally leads to a corollary: the velocity of the slow electron holes will eventually stabilize at the local minimum of the ion VDF[38,43]. On the other hand, the slow electron holes in the accelerated (or decelerated) state will, in turn, change the ambient ion VDF: when an ion passes through the electron hole parallel (anti-parallel) to the acceleration direction of the electron hole, the ion is accelerated (decelerated). Thus, slow electron holes are not simply controlled by the kinetic features of the ion VDF but are coupled with each other to reach a final stable equilibrium state.

## Results

Several slow electron holes were observed by MMS satellites on May 28, 2017 (Fig. 2), when the four MMS spacecraft were located in the plasma sheet at the Geocentric Solar Ecliptic (GSE) coordinates (−19, −11, 3)$R_E$. The local magnetic field is about (4.5, 8.6, 2.2) nT in GSE coordinates and the plasma density detected by MMS is below 0.1 cm$^{-3}$. There are four bipolar parallel electric field structures A–D highlighted in cyan shading, of which the first three holes are clustered together and the fourth hole is far apart (Fig. 2a). In this event, the perpendicular electric field of the electron holes is much smaller than the parallel electric field, due to the fact that the electron holes have the spatial width in the plane perpendicular to the local magnetic field much larger than the parallel spatial width[6,37,44]. An ion beam drifts anti-parallel to the local magnetic field (Fig. 2b), which is often observed together with slow electron holes[38]. The bipolar structures in the parallel electric field are observed by all the four MMS satellites in the order of MMS1-MMS3-MMS4-MMS2 (Fig. 2c). We use the four-spacecraft timing method to estimate the drift velocity of this

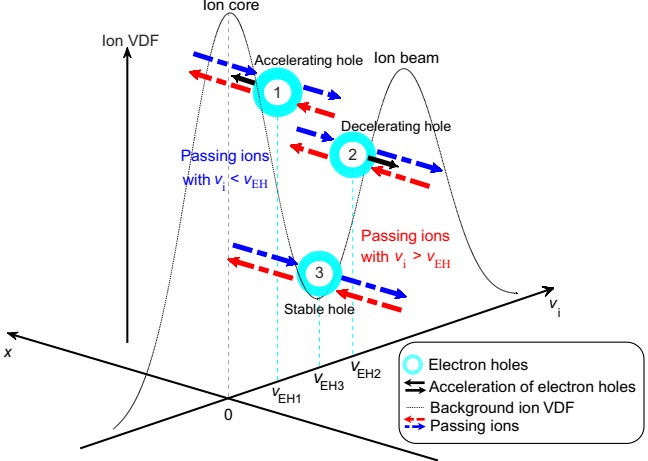

**Fig. 1 | Schematic illustration of the interaction of slow electron holes with local ions with a double-humped velocity distribution.** The black double-humped dotted line indicates the background ion velocity distribution function (VDF), which is a combination of ion core and beam populations described by Maxwell distributions. Cyan hollow spheres represent three electron holes with different velocities ($v_{EH_1}$, $v_{EH_2}$ and $v_{EH_3}$). The holes drift along the $x$ axis, and the black arrows indicate their acceleration. Hole 1 is accelerating due to its velocity located at the negative gradient of the ion VDF, whereas hole 2 is decelerating because its velocity is at the positive gradient. Hole 3 is stable because its velocity is right at the local minimum of the ion VDF. Dotted arrows indicate passing ions, red and blue indicate traversing along the $x$ or $-x$ axis, respectively. The length of the arrow indicates the velocity of passing ions.

solitary wave as $v_{EH} \approx 996$ km s$^{-1}$ (labeled by C in Fig. 2a), and the propagation angle is about 175° in the spacecraft frame (see Methods section, Eq. (1) in Multispacecraft timing method for details). The same method is used to estimate the drift velocities and propagation angles of other three solitary waves (labeled by A, B, D in Fig. 2a). These three solitary waves all drift almost anti-parallel (>174°) to the local magnetic field at velocities of 800–1000 km s$^{-1}$, slightly greater than the local ion thermal speed ($v_{thi} \approx 490$ km s$^{-1}$) and slower than the ion beam but only a few percent of the local electron thermal speed ($v_{the} \approx 3.4 \times 10^4$ km s$^{-1}$). Thus, we interpret these positive potential solitary structures as slow electron holes drifting almost anti-parallel to the magnetic field, consistent with the observed occurrence of the slow electron holes aboard four MMS spacecraft.

The spatial separations of the MMS satellites along the magnetic field (Fig. 2d) and the observation time delays of the slow electron holes (Fig. 2e) on each pair of satellites can be used to estimate the velocity of the slow electron holes using two-spacecraft interferometry. The four-spacecraft timing method has certain limitations, since it does not allow us to estimate the acceleration/deceleration rate of the hole. The two-spacecraft interferometry is more direct, and allows estimating the acceleration/deceleration rate, but has to presume either strictly parallel or anti-parallel propagation to the local magnetic field. Compared with the four-spacecraft timing analysis method, the two-spacecraft interferometry can obtain six velocity estimates using six pairs of MMS spacecraft. Assuming that the slow electron holes propagate strictly anti-parallel to the local magnetic field, we use two-spacecraft interferometry to estimate the drift velocity of the slow electron hole C in Fig. 2a between different adjacent satellites. The three velocity estimates of this slow electron hole in Fig. 2c corresponding to MMS1-MMS3, MMS1-MMS4, and MMS1-MMS2 are −1250, −1077, and −982 km s$^{-1}$, respectively, which indicates that drift velocity of this slow electron hole keeps decreasing, rather than drifting at a constant velocity. The other three velocity estimates of this slow electron hole also support this conclusion.

In the four-spacecraft timing method, we presume that the electron hole is moving at a constant velocity and direction. But if the electron hole is accelerating/decelerating, the velocity and direction obtained by the four-spacecraft timing method are not accurate enough. Therefore, in the following calculation and analysis, we assume that all electron holes drift strictly anti-parallel to the local magnetic field and have a fixed acceleration/deceleration rate over the detected period. We use simultaneous observations from the four MMS satellites to estimate the acceleration rate of the electron holes. The four sets of spatial separations (Fig. 2d) and time delays (Fig. 2e) as position-time ($x - t$) data are fitted to the equation of uniformly accelerating motion $x = U_0 t + \dot{U} t^2 / 2$ ($\dot{U}$ is the acceleration rate and $U_0$ is the initial velocity) to estimate the acceleration rate of the electron holes. The MMS spacecraft moves at a speed less than 1 km s$^{-1}$, and thus its speed can be neglected in comparison to the speed of the electron holes. Therefore, we regard the spacecraft frame approximately as the inertial frame.

Since previous theoretical analysis showed that the acceleration/deceleration rate of the electron hole is related to the gradient of the local ion VDF at the speed of the electron hole[43], we try to find the relationship between the fitted acceleration/deceleration rate of the four slow electron holes and the gradient of the ion VDF at the corresponding slow electron hole velocity. Since the velocities of all the four electron holes are negative, the fitted acceleration/deceleration rates for holes C and D are positive, indicating that they are decelerating (Fig. 3c, d), and their velocities are at the positive gradients of the ion VDF (Fig. 3g, h). The hole B is accelerating due to a negative acceleration (Fig. 3b), which is consistent with a negative gradient of the ion VDF at the velocity of hole B (Fig. 3f). Note that the fitted acceleration error of the hole A using the four MMS satellites is large, and the R-squared coefficient is smaller than those for the other

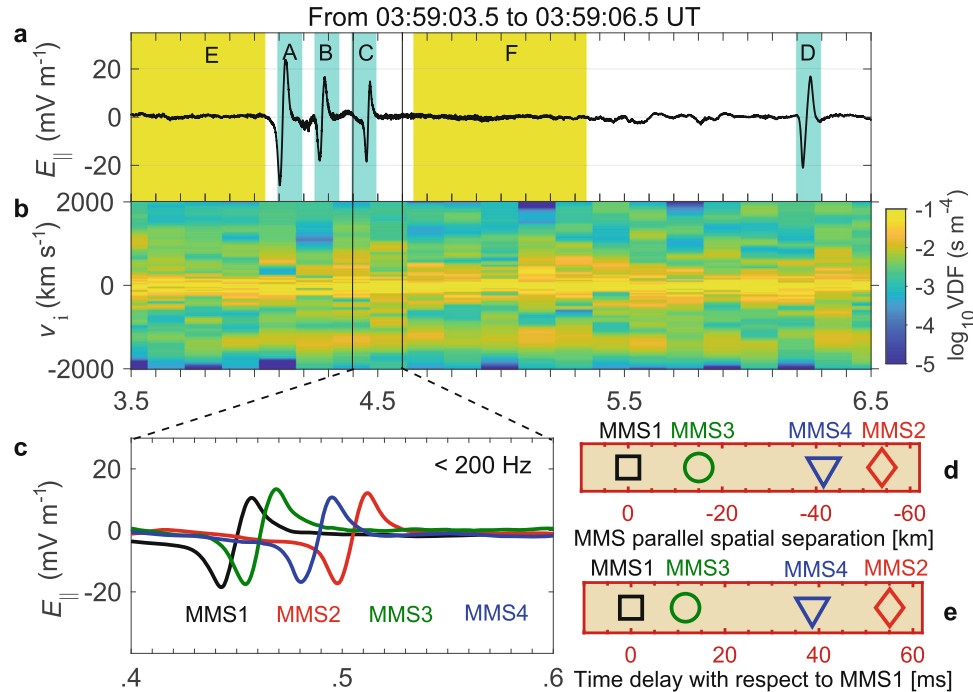

**Fig. 2 | Magnetospheric Multiscale observations in the plasma sheet. a** Parallel electric field measured at 8192 samples s$^{-1}$ resolution by the electric double probe[49, 50]. The cyan shaded regions A–D represent the four detected electron holes. **b** Ion one-dimensional velocity distribution function (VDF) (integrated from the ion three-dimensional distribution, measured at 150 ms cadence by the fast plasma investigation instrument[51]), and the *Y*-axis is the ion parallel velocity. The ion one-dimensional VDF has a significant enhancement near −1500 km s$^{-1}$ after 03:59:03.8 UT and is separated from the ion core (speed below 500 km s$^{-1}$), which is identified as an anti-parallel ion beam. **c** Low pass (<200 Hz) parallel electric field waveform of the solitary wave C. **d** Spatial separations between the Magnetospheric Multiscale (MMS) spacecraft along the local magnetic field. **e** Time delay between observations of the solitary wave C.

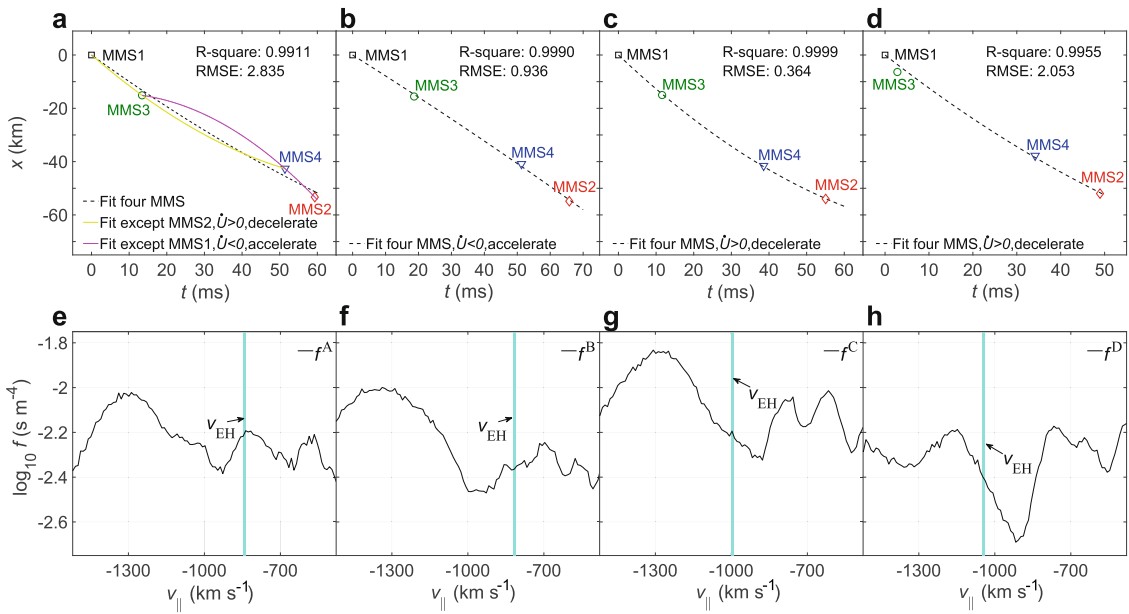

**Fig. 3 | Acceleration fitting and ion distribution around the electron hole velocity. a–d** Fitting motion curves of the four electron holes A–D obtained based on space separations and time delays of the Magnetospheric Multiscale (MMS) satellites. The yellow curve in **a** denotes the motion curve of hole A calculated with MMS1-MMS3-MMS4, and the magenta curve represents the motion curve of hole A calculated with MMS3-MMS4-MMS2. The black dotted curves in **a–d** denote the fitted motion curves of holes A–D. Note that positive acceleration rate indicates that the hole is slowing down (holes C and D) and negative acceleration rate means that the hole is speeding up (hole B), since the holes drift anti-parallel to the magnetic field. The estimated acceleration/deceleration rate $\dot{U}$ of the four electron holes A–D are $5.7 \times 10^3$, $-1.5 \times 10^3$, $1.3 \times 10^4$ and $9.6 \times 10^3$ km s$^{-2}$, which are supported by theoretical predictions (see Supplementary Fig. 2 and Supplementary Methods: Theoretical estimates of the acceleration/deceleration rates for details). **e–h** Local ion velocity distribution function around the $v_{\mathrm{EH}}$ in regions A–D, respectively. The cyan vertical lines in **e–h** indicate the average velocity of the corresponding hole across all spacecraft.

three electron holes (Fig. 3a), meaning that the acceleration of the hole A may change drastically during the passage across four MMS satellites. So, we use the first three satellites (MMS1-MMS3-MMS4) and the last three satellites (MMS3-MMS4-MMS2) to obtain two estimates of the acceleration rate, resulting in a positive value and negative one, respectively. In fact, the velocity of the hole A is at the peak of the local ion VDF (Fig. 3e), meaning that the complex and variable gradient direction of the ion VDF should be the cause of its unstable acceleration (first deceleration and then acceleration). Therefore, the acceleration of the electron hole depends on the local ion VDF gradient at the velocity of the electron holes ($v_{EH}$). When $v_{EH}$ is at the positive (negative) gradient of the ion VDF, the electron holes are decelerated (accelerated). Eventually, the velocity of the electron hole is expected to stabilize at the local minimum of the ion VDF (see Supplementary Fig. 1 and Supplementary Discussion for details), which is supported by the theory of electron hole kinematics and statistical observations[38,43]. Our result provides an observational bridge between the kinetic features of the ion distribution function and the acceleration of the slow electron holes.

We perform theoretical calculations based on actual plasma parameters ($L_{||}$, $\dot{U}$ and $\Phi$, which are the parallel scale, acceleration/deceleration rate and central potential of an electron hole, respectively) and attempt to find the evidence from observations that accelerating (or decelerating) slow electron holes affect ion VDF (see Methods section, Eq. (9) in Transit time and velocity difference of ions passing through electron holes). The ion transit time ($\delta t$) across a single electron hole is on the order of tens of ms (Fig. 4a), and the velocity difference ($\Delta v'$) in the inertial frame is several km s$^{-1}$ (Fig. 4b). Basic parameters are set to $L_{||0} = 5$ km, $\dot{U}_0 = 10^4$ km s$^{-2}$, and $\Phi_0 = 300$ V. The transit time $\delta t$ is affected by the $L_{||}$ and $\dot{U}$ of the hole. The larger $L_{||}$ or smaller $\dot{U}$ is, the longer the $\delta t$ will be. $\Phi$ has little influence on $\delta t$, so it is not illustrated in the figure. For a constant $v_1$, $\Delta v'$ is positively correlated with $L_{||}$, $\dot{U}$, and $\Phi$, of which $\Phi$ has the most significant effect. The $\Delta v'$ of the passing ions ranges from a few km s$^{-1}$ to tens of km s$^{-1}$, about several percent of the $v_{thi}$, The drift velocity ($v_{EH} = 841$, 817, and 996 km s$^{-1}$) of the holes A−C are estimated by the four-spacecraft timing method. The central potential ($\Phi = 360$, 260, and 230 V) is the maximum of $\Phi = \int E_{||} v_{EH} dt$. $E_{||}$ is the parallel electric field. $L_{||}$ of the three holes are 9.8, 8.4, and

7.0 km, which are derived from $L_{||} = v_{EH} t_{pp}/2$, where $t_{pp}$ is the time difference between the peak and valley of the electron hole $E_{||}$ detected by the MMS satellites. The local Debye length is 2.5 km, and the electron and proton gyroradii are 29 km and 733 km, respectively. So, the parallel scales of electron holes are several Debye lengths and much smaller than the electron and proton gyroradii. The actual central potential and parallel scale may be larger than the estimated values since the satellites did not necessarily pass through the center of the hole. High-speed anti-parallel ions (1500–2000 km s$^{-1}$) in region E are considered to originate in region F, pass through three electron holes and finally reach the region E (Fig. 2a and Fig. 4c). Based on the fitted acceleration/deceleration rates of the three holes A−C, according to the theoretical calculation (see Methods section, Eq. (9) in Transit time and velocity difference of ions passing through electron holes for details), for the passing ions (from −1500 to −2000 km s$^{-1}$) in Fig. 4c, the velocity difference $\Delta v'$ is tens of km s$^{-1}$ along the local magnetic field (Fig. 4b). The theoretical curve in Fig. 4c is the mapping of the ion distribution (for passing ions, from −1500 km s$^{-1}$ to −2000 km s$^{-1}$) from region F to region E, which is in agreement with the ion VDF observed aboard MMS satellites and further supports our conclusion. Note that there are significantly more reflected ions in region F than those in region E (blue box in Fig. 4c), plausibly because the ions in region F are reflected by the positive potential of electron holes (C and D) on both sides, thus causing ion accumulation. The parallel E-field waveforms detected by satellites may not be strictly symmetrical because of asymmetric ion reflection. After a careful analysis, we find that the net potential difference across the electron holes caused by a slight asymmetry is only a few volts, which has little effect on the velocity of passing ions (velocity range is from −1500 to −2000 km s$^{-1}$). So, the passing ion acceleration caused by this net potential difference across the electron holes is negligible. Thus, the combination of theoretical calculations and observations has demonstrated the effect of electron holes with acceleration $\dot{U}$ on the net velocity of ions passing through them. Similarly, accelerating (or decelerating) electron holes will also change the net speed of electrons passing through them. However, the acceleration effect on electrons is just opposite to that of ions, because electrons are negatively charged.

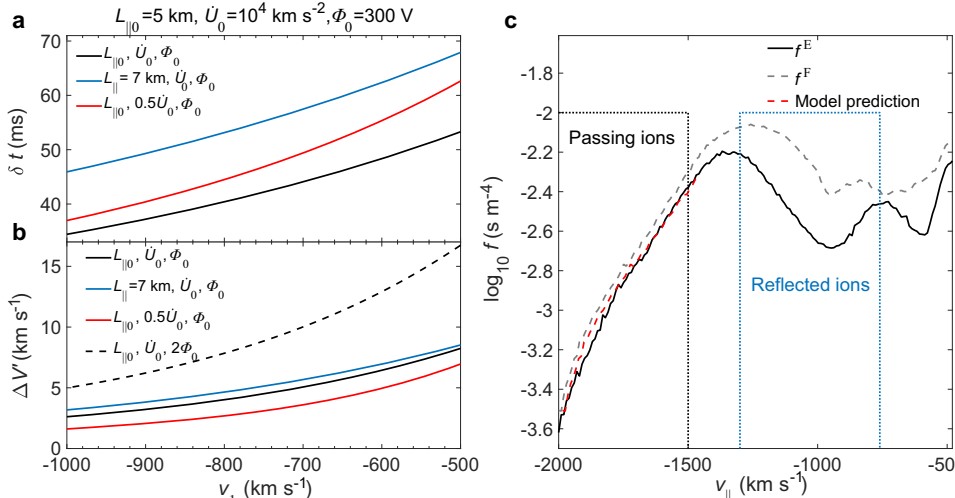

**Fig. 4 | Velocity change of ions passing through the electron holes. a, b** Transit time $\delta t$ and velocity difference $\Delta v'$ of ions passing through the electron holes for several sets of parameters ($L_{||}$, $\dot{U}$ and $\Phi$: the parallel scale, acceleration/deceleration rate and central potential of the electron holes). **c** The solid and dashed black lines represent ion one-dimensional velocity distribution function in regions E and

F, respectively. Regions E and F are the yellow shaded areas in Fig. 2a. The red dotted line represents theoretical prediction. The velocity distribution function of passing ions (from −1500 to −2000 km s$^{-1}$) and reflected ions (from −760 to −1300 km s$^{-1}$) are in the black and blue boxes, respectively.

## Discussion

We take advantage of the simultaneous multi-satellite measurements and high-resolution data of the MMS satellites to find the accelerating (or decelerating) electron holes, and further observationally demonstrate that the acceleration of slow electron holes depends on the gradient of the local ion VDF at the drift velocity of electron holes. Combining theoretical calculations with satellite observations, we find that slow electron holes with non-zero acceleration can cause the net velocity change of ions passing through them. In the frame of electron holes, when an ion passes through the electron hole in the direction parallel (anti-parallel) to the acceleration of electron holes, the ion is accelerated (decelerated). Therefore, in the hole frame, through electron holes with non-zero acceleration, the energies of ions with relative velocity direction (faster or slower than the electron holes) parallel to the acceleration are exchanged from those in the anti-parallel direction, which provide an energy exchange channel in collisionless plasma. The effect of the background ion velocity distribution gradient at the electron hole velocity also arouses our thinking on the origin of the slow electron hole. Previous studies have suggested that slow electron holes and fast electron holes originate from different plasma instabilities[5,7,17,27,45–47]. Positive ion VDF gradient provided by the ion beam can prevent electron holes from self-accelerating into fast electron holes and maintain their speed near the ion thermal speed, thus being identified as slow electron holes[38]. Therefore, the ion beam is an important factor to maintain the existence of slow electron holes. Since the interaction between electron holes and ion beams is a universal physical process, the results presented here also have important implications for understanding the evolution of plasma distributions and associated microscopic interactions in space plasma.

## Methods

### Multispacecraft timing method

We use Multispacecraft timing method[48] to estimate the drift velocity of electron holes in the spacecraft frame. Let us assume that an electron hole is moving in the direction $\mathbf{n}$ with velocity $V$. In the simplest case, the electron hole can be identified unambiguously on all the four spacecraft. The structure is observed at time $t_\alpha$ by the spacecraft $\alpha$, $1 \leq \alpha \leq 4$, which is located at position $\mathbf{r}_\alpha$. Let us arbitrarily take spacecraft 1 as the reference. During the time $t_\alpha - t_1$ the structure moves along the normal direction a distance $V(t_\alpha - t_1)$, which is equal to the projection of the separation distance $\mathbf{r}_\alpha - \mathbf{r}_1$ onto $\mathbf{n}$,

$$\left(\mathbf{r}_\alpha - \mathbf{r}_1\right) \cdot \mathbf{n} = V\left(t_\alpha - t_1\right) \tag{1}$$

Equation (1) can be rewritten as

$$D\,\mathbf{m} = \mathbf{T} \tag{2}$$

where $\mathbf{m}$ is a vector representing $\mathbf{n}$ over $V$

$$\mathbf{m} = \frac{\mathbf{n}}{V} \tag{3}$$

D is the $3 \times 3$ matrix defined by

$$D = \left(\mathbf{r}_2 - \mathbf{r}_1, \mathbf{r}_3 - \mathbf{r}_1, \mathbf{r}_4 - \mathbf{r}_1\right) \tag{4}$$

and $\mathbf{T}$ is a linear array

$$\mathbf{T} = \begin{pmatrix} t_2 - t_1 \\ t_3 - t_1 \\ t_4 - t_1 \end{pmatrix} \tag{5}$$

Hence

$$\mathbf{m} = D^{-1}\mathbf{T} \tag{6}$$

It is easy to determine both $\mathbf{n}$ and $V$, and get the propagation angle of the electron hole. Note that a necessary condition for a solution is that the four spacecraft are not coplanar. Due to the tetrahedral structure of the MMS satellites, the four satellites will never be coplanar, so there is always a solution $\mathbf{n}$ and $V$.

### Transit time and velocity difference of ions passing through electron holes

We adopt part of the theoretical method proposed by Hutchinson[36] in calculating the total change rate of the ion stream's momentum due to the hole's presence and acceleration to calculate the velocity change of the ion passing the hole. In the accelerating hole frame, ions are subjected to an additional inertial force, so that

$$m_i \frac{dv}{dt} = -q\frac{\partial \Phi}{\partial x} - m_i \dot{U} \tag{7}$$

where $m_i$ is the mass of a proton, $\Phi(x)$ is the potential of the electron hole, and $q = +e$ is the charge of the proton. For constant $\dot{U}$, in the accelerating hole frame, the velocity change of ions entering and leaving the electron hole can be obtained by integrating the Eq. (7)

$$v_2^2 - v_1^2 = -2\dot{U}(x_2 - x_1) \tag{8}$$

Note that $x_1$ and $x_2$ are the positions of the beginning and the end of the electron hole structure, which are constants in the hole frame. Other quantities with subscripts 1 or 2 are naturally values at $x_1$ or $x_2$, especially $\Phi_1 = \Phi_2 = 0$. Therefore, the velocity difference in the accelerating hole frame is only related to the acceleration and parallel scale of the electron hole structure. When calculating the velocity difference in the inertial frame, the velocity difference of the hole during the ion transit across is additionally added. We use $v$ and $v'$ to distinguish whether the velocities are in the hole or inertial frames. Consequently, combined with Eq. (8), the velocity difference in the inertial frame is

$$v_2' - v_1' = v_2 - v_1 + \delta t \dot{U} = \sqrt{v_1^2 - 2\dot{U}\left(x_2 - x_1\right)} - v_1 + \dot{U}\int_{x_1}^{x_2}\frac{dx}{v} \tag{9}$$

where

$$\delta t = t_2 - t_1 = \int_{x_1}^{x_2}\frac{dx}{v} \tag{10}$$

is the transit time across the hole of an ion, and

$$v(x) = \begin{cases} \sqrt{v_1^2 - \frac{2q\Phi(x)}{m_i} - 2\dot{U}(x - x_1)}, & v_1 > 0 \\ -\sqrt{v_1^2 - \frac{2q\Phi(x)}{m_i} - 2\dot{U}(x - x_1)}, & v_1 < 0 \end{cases} \tag{11}$$

is the velocity of an ion (entering the hole at $x_1$) in the hole frame. Note that $v_1$ is the initial velocity of the ion in the hole frame, which is an independent quantity. Therefore, the velocity difference in the inertial frame is related to the hole acceleration/deceleration rate ($\dot{U}$), the parallel length scale ($L_\parallel$) of the hole, and the peak amplitude of the electrostatic potential ($\Phi$). These parameters ($\dot{U}$, $L_\parallel$, and $\Phi$) can all be estimated according to the observation data provided by the satellites.

For an ion with an initial velocity of $v_1$ passing through an electron hole with an initial speed of $v_{EH}$, its velocity in the inertial frame is

$$v'(x) = v(x) + \dot{U} \int_{x_1}^{x} \frac{dx}{v(x)} + v_{EH} \qquad (12)$$

## Data availability

MMS observations are available via the Science Data Center at CU/LASP (https://lasp.colorado.edu/mms/sdc/public/data/). The datasets generated during and/or analyzed in the current study are available from the corresponding author upon request. Source data are provided with this paper.

## Code availability

MMS data are loaded, analyzed, and plotted using the MATLAB software with the irfu-matlab code package, which can be downloaded from https://github.com/irfu/irfu-matlab.git.

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

## Acknowledgements

We acknowledge the entire Magnetospheric Multiscale (MMS) satellite team for providing the excellent data. We are grateful to I. H. Hutchinson for theoretical discussions of electron holes. This work is supported by the National Natural Science Foundation of China (41925018, 41874194).

## Author contributions

Z.Y. led this study and supervised the project development. Z.Y. and Y.D. conceived the idea and interpreted the event. The manuscript is largely written by Y.D. with contributions from Z.Y., Y.D. fitted the acceleration of electron holes and calculated the velocity change of ions passing through electron holes. S.H., Z.X., and X.Y. contributed to the data analysis and revised the manuscript. C.J.P., R.B.T., and J.L.B. contributed to data processing. All authors reviewed and approved the final manuscript.

## Competing interests

The authors declare no competing interests.
