## [Peer Review File · Nature Communications]

REVIEWER COMMENTS

Reviewer #1 (Remarks to the Author):

In the manuscript “Can accelerating slow electron holes exist and cause the net velocity change of ions passing through them?” the Authors provide some evidence for acceleration/deceleration of slow electron holes and associated with that process variation of the momentum of ions passing through the electron holes. This study is rather interesting, since that is potentially the first report of accelerating/decelerating electron holes in space plasma. However, first of all I believe the results are of interest for only a narrow community, whose research is focused on solitary waves. The Authors claim in lines 26-28 that “The results have important implications for understanding the evolution of plasma distributions and associated microscopic nonlinear interactions at magnetized planets throughout the solar system and beyond”, but this statement is too loose and general to stimulate an interest of a wider community. The solitary waves are indeed universal structures that have been observed in various space plasma regions, but the Authors’ results are more important from the general plasma physics perspective rather than for understanding of “microscopic nonlinear interaction at magnetized planets”, where many other processes (depending on a planet and its surroundings) play a far more critical role. Even more important is that the methodology used by the Authors is far from being fully justified and in several places is not coherent and even self-contradictory. Thus, the publication of this manuscript in a journal like Nature oriented for a wide community does not seem to be well justified to me, since the significance of the presented results for a wider scientific community has not been clarified. Also taking into account that methodology used by the Authors has some peculiar problems, I believe this manuscript is better to be submitted to a more specialized journal (e.g., Journal of Geophysical Research). Also the English of the manuscript is far from being perfect and needs substantial revisions. My major and some of the minor comments are listed below.

Major Comments

1. The Authors obtain the electron hole acceleration experimentally, but do not compare the derived acceleration/deceleration rates to theoretical predictions. The analysis by Hutchinson, 2021, <https://link.aps.org/doi/10.1103/PhysRevE.104.015208> predicts the acceleration rate depending on the slope of the ion distribution function. Since only a few electron holes have been considered, the Authors need to demonstrate that the inferred acceleration/deceleration and its correlation with the ion VDF slope is actually not a lucky coincidence by (1) comparing their observations to the theoretical predictions and (2) expanding the dataset up to hundreds of events and support the results statistically. I strongly recommend doing both (1) and (2), since this potentially novel effect needs to be carefully tested. Demonstrating the effect for just four events is not a solid evidence; moreover, the Authors can easily get at least a few hundred similar events (I checked Ref. 38 cited by the Authors and can see more than a thousand electron holes considered there).

2. Line 140: The Authors compare only the distributions of ions with velocities of 1,500-2,000 km/s streaming from region F to region E and claim to identify the velocity variation by a few tens of km/s consistent with theory predictions. However, the distribution functions in regions F and E differ by more than a few tens of km/s at other velocities and this difference does not seem to be consistent with theory. In particular, there are ions propagating from region E to region F, and they should be decelerated according to theory. Do the Authors observe the corresponding effect? It seems that this effect is not observed and, therefore, the Authors' computations do not entirely support the observations. I strongly recommend to do the mapping of the ion distributions from F to E and vice versa and carry out a more careful comparison between theory and observations. Otherwise the comparison between theory and observations is far from being complete.

3. Line 70-71 and 79-82: Using four spacecraft observations the Authors find that the electron hole propagates at 175 deg to local magnetic field at the speed of about 996 km/s. Later in lines 79-82 the Authors assume that electron holes actually propagate strictly parallel to local magnetic field and obtain other speed estimates and infer acceleration/deceleration rates. However what if the electron hole indeed propagates at 175 deg to local magnetic field? In this case the two-spacecraft interferometry applied to whatever pair of MMS spacecraft would provide the speed of 996 km/s and no acceleration/deceleration would be obtained. Thus, there is a fundamental inconsistency in the Authors' analysis and their estimates of electron hole acceleration/deceleration rates are strongly dependent on the electron hole propagation direction. The deviation by only 5 deg off the strictly parallel propagation makes acceleration/deceleration rates equal zero!

4. Line 75: "the local ion thermal velocity" – Did the Authors compute the thermal speed for the entire ion distribution function? If yes, the distribution is highly non-Maxwellian and, thus, the concept of temperature is inapplicable. The thermal speed of the ion core population would be something reasonable to use in this case.

Minor comments

--- Line 18: "with acceleration" – Should it be "acceleration/deceleration"? Some of the electron holes in Figure 2 are decelerating.

--- Line 18: "with acceleration on basis of" is better to reformulate as "whose acceleration is revealed using"

--- Line 21: "detection time delay of the MMS satellites" should be "detection time delays between different MMS satellites"

--- Line 23: "through the decelerating electron holes" – Now I am confused. Are the electron holes accelerating or decelerating? A sentence above the Authors were calling their electron holes "accelerating".

--- Line 23: ", which are consistent with the theoretical predicted results" should be "in accordance with theoretical predictions"

--- Line 25: "ions passing through them in the same direction as the acceleration of electron holes? – What about the ions passing in the opposite direction?"

--- Line 45; "electron holes may accelerate and decelerate due to the kinetic features of the ion VDF" - The correct reference is Ref [43].

--- Line 52: "the slow electron hole" should be "slow electron holes"

--- Line 54: "theory analysis" should be "theory"

--- Line 55: "due to the kinetic features" should be "depending on kinetic features"

--- Line 56: "accelerations of electron holes have" should be "acceleration of electron holes has"

--- Line 60: "the net velocity" should be "a net velocity"

--- Line 64: "the plasma sheet" – More information needed to clarify where exactly the spacecraft were located, what the background plasma conditions were etc.

--- Line 78: "by four MMS spacecraft." should be "at four MMS spacecraft"

--- Line 83-84: "can obtain three estimated velocities based on four MMS observations" – One can actually get six different pairs of MMS spacecraft. Thus the Author should add three more estimates to their analysis. Does this change the Authors' results?

--- Line 84: "antiparallely drifts" should be reformulated.

--- Line 88: "decaying" is not a good term here, a better term would be "increasing"

--- Line 94: "in the inertial frame is U" – What "inertial frame" do the Authors mean here? What is "U"? It has not been defined prior to Line 94.

--- I cannot see in Figure 2 or anywhere before Line 121, where the analysis was done, anything about the acceleration/deceleration rates.

--- Line 114: "changeable" - A better terms is probably "variable"

--- Line 141-143: The text in these lines is difficult to understand, English must be improved.

--- Line 77: "the antiparallel direction of the magnetic field" - better to say "anti-parallel to the magnetic field"

--- Line 87: "MMS1-MMS3, MMS1-MMS4, and MMS1-MMS2 are" - There is actually six pairs among four spacecraft. Thus the Authors can actually use three more pairs to see whether the velocity increase/decrease is confirmed by the other pairs of spacecraft.

--- Line 173 "In the frame of ion holes" is likely a misprint.

"to the accelerations" should be "to the acceleration"

--- Line 179: "Previous studies have suggested that slow electron holes and fast electron holes originate from different plasma instabilities^{5,7,17,27,47,48,49}. However, the positive ion VDF gradient provided by the ion beam can prevent electron holes from self-accelerating into fast electron holes and maintain their speed near the ion thermal velocity, thus being identified as slow electron holes." - I don't see a statement of these two sentences. Yes, there are solid arguments that slow and fast electron holes are produced by different mechanisms, and that does not contradict the second sentence that slow electron holes can remain slow if the distribution function has a local minimum and the electron hole velocity resides near that minimum without self-acceleration. What is the point of "However" in the beginning of the second sentence and what is the whole point of these two sentences?

Reviewer #2 (Remarks to the Author):

The authors present observations of slow electron phase space holes (EHs) from MMS which include the presence of a passing ion beam. Using multi-spacecraft timing analysis, the EHs are found to be accelerating rather than moving at constant speed. A symmetric EH potential with changing speed has been theorized as capable of accelerating the passing ion population, which appears to be consistent with changes in the ion phase-space density distribution. This is a unique finding which has been investigated thoroughly by the authors and will be useful to others in the field. The methodology is sound and detailed. However, organizational problems and a few notes of clarification hold the paper back from being ready for publication. Most of these may be possible to address in the editing process after minor revisions.

The manuscript – primarily the introduction – is fraught with grammatical errors. None of these directly impact the scientific content, but a corrections pass would be highly beneficial.

Organization:

Some of the organizational issues come from Nature Communications' "methods section last" format, not the author's design. Even so, references to the methods section are not clear, and make it difficult to parse what was actually done. References via equation numbers or other description at first mention of each method/variable in the text are needed.

1. Multispacecraft timing method (e.g. line 70, 134). This method is straightforward, but requires a citation or more thorough description. For example, it is not explained anywhere where the propagation angle comes from.
2. "Theoretical calculation" (e.g. line 143). This is well-explained in the methods section, but at this stage is unclear.
3. EH parameters L , U_{dot} , ϕ are used in line 122, but have not yet been defined.
4. Likewise, primed and unprimed velocities are not defined outside of the methods section. For both points 3 and 4, a mention of "see Methods..." is used, but it is still unclear what the reader should look for.

The text of the discussion section serves as more of an introduction/background, and would be much more useful before describing the results. Figure 4 is excellent.

Other comments:

Lines 147-148 and conclusions:

It is difficult to say whether deceleration of ions has really been observed. The distribution could just as easily have decreased in density overall. Indeed, looking at the peak of the distribution, it has actually shifted left a bit, suggesting acceleration (if not for the complication of it being outside the 'passing' region, and likely including reflected ion populations). A more nuanced interpretation and conclusion is needed here.

Lines 137-139: Is there a perpendicular component to the E-field? The EHs seem quasi-1D in this instance.

By eye, the EHs in this event appear to be nearly symmetric, but the formula used for ion acceleration assumes perfect symmetry. Is there a slight asymmetry in the data? And how would ion acceleration by that potential compare to the theorized ion acceleration due to the EH motion?

In the results section, some indication of the ion/electron Debye length and gyro-radius to compare with the EH scales would be useful.

Reviewer #3 (Remarks to the Author):

This paper uses multi spacecraft MMS observations of symmetric potential electron holes in the magnetotail to verify the theoretical prediction that electron holes with non-zero acceleration can accelerate ions passing through them. Using MMS high resolution data, they investigate the ion distribution functions in these 4 slow electron holes that are decelerating and they show that ions velocities are changing through these holes.

This paper is well written and easy to follow. I will recommend it for publication after authors address my comments.

Major Comment:

Lines 66-68: It is not clear to me how you conclude there is a beam drifting antiparallel to magnetic field from Figure 1b? Could you please explain the figure? Is the beam speed around -1500km/s?

Line 74: So technically they are drifting with the ion beam or close to the ion beam speed?

Minor Comments:

- Line 78: consistent with the observed ...

Response Letter to Reviewers

We are grateful for the constructive comments on this manuscript (NCOMMS-23-26818-T) from all the reviewers. In the text below, each reviewer comment is quoted in italics and is followed by the corresponding detailed response. We have also carefully revised the manuscript and added the Supplementary Information. These updates are highlighted in red in those files. In the text below, the references to these updates are highlighted in a similar way.

GENERAL COMMENTS FROM REVIEWER #1:

In the manuscript “Can accelerating slow electron holes exist and cause the net velocity change of ions passing through them?” the Authors provide some evidence for acceleration/deceleration of slow electron holes and associated with that process variation of the momentum of ions passing through the electron holes. This study is rather interesting, since that is potentially the first report of accelerating/decelerating electron holes in space plasma. However, first of all I believe the results are of interest for only a narrow community, whose research is focused on solitary waves. The Authors claim in lines 26-28 that “The results have important implications for understanding the evolution of plasma distributions and associated microscopic nonlinear interactions at magnetized planets throughout the solar system and beyond”, but this statement is too loose and general to stimulate an interest of a wider community. The solitary waves are indeed universal structures that have been observed in various space plasma regions, but the Authors’ results are more important from the general plasma physics perspective rather than for understanding of “microscopic nonlinear interaction at magnetized planets”, where many other processes (depending on a planet and its surroundings) play a far more critical role. Even more important is that the methodology used by the Authors is far from being fully justified and in several places is not coherent and even self-contradictory. Thus, the publication of this manuscript in a journal like Nature oriented for a wide community does not seem to be well justified to me, since the significance of the presented results for a wider scientific community has not been clarified. Also taking into account that methodology used by the Authors has some peculiar problems, I believe this manuscript is better to be submitted to a more specialized journal (e.g., Journal of Geophysical Research). Also the English of the manuscript is far from being perfect and needs substantial revisions. My major and some of

the minor comments are listed below.

Response from Authors:

Thank you for your decision and constructive comments on my manuscript. We also thank the reviewer for the positive and encouraging comments, especially that “*This study is rather interesting, since that is potentially the first report of accelerating/decelerating electron holes in space plasma.*”. We have carefully considered the suggestion of Reviewer and make some changes in the revised manuscript.

In the following, we address the specific comments point-by-point.

SPECIFIC COMMENTS FROM REVIEWER #1:

Reviewer #1 -- Major Comment 1:

The Authors obtain the electron hole acceleration experimentally, but do not compare the derived acceleration/deceleration rates to theoretical predictions. The analysis by Hutchinson, 2021, <https://link.aps.org/doi/10.1103/PhysRevE.104.015208> predicts the acceleration rate depending on the slope of the ion distribution function. Since only a few electron holes have been considered, the Authors need to demonstrate that the inferred acceleration/deceleration and its correlation with the ion VDF slope is actually not a lucky coincidence by (1) comparing their observations to the theoretical predictions and (2) expanding the dataset up to hundreds of events and support the results statistically. I strongly recommend doing both (1) and (2), since this potentially novel effect needs to be carefully tested. Demonstrating the effect for just four events is not a solid evidence; moreover, the Authors can easily get at least a few hundred similar events (I checked Ref. 38 cited by the Authors and can see more than a thousand electron holes considered there).

Response from Authors:

We thank the reviewer for raising this issue, which indeed was not sufficiently addressed in the original manuscript.

We thank the reviewers for two instructive suggestions to enhance the reliability of our conclusions.

For the first suggestion “*comparing their observations to the theoretical predictions.*”.

Comparing the derived acceleration/deceleration rates to theoretical predictions is indeed a final solution to this issue, but quantitative comparisons are difficult to make with current theories and observations. On the one hand, the analysis by Hutchinson, 2021 indeed reach conclusions “*It is*

shown that to avoid the self-acceleration of the hole velocity away from ion velocities it must lie within a local minimum in the ion velocity distribution. Quantitative criteria for the existence of stable equilibria are obtained.” and “Long-lived one-dimensionally stable slow electron hole equilibria can exist only when the background ion velocity distribution has a sufficiently deep local minimum and the electron hole speed lies within it.”. However, **there is no quantitative analysis of the correlation between the acceleration/deceleration rate of electron hole and the slope of ion VDF.** On the other hand, **the slope of the ion VDF is difficult to determine precisely.** The resolution of the ion sampling data of the MMS spacecraft is 150 ms, and the time scale for the satellite to detect an electron hole is only tens of ms, so the slope estimation error is large. In summary, the current theoretical and observational basis is not sufficient to achieve this quantitative comparison. Instead, the qualitative comparison between the acceleration/deceleration rate of electron hole and the slope of ion VDF is reported in our manuscript.

For the second suggestion *“expanding the dataset up to hundreds of events and support the results statistically.”.*

Slow EHs are indeed common in space plasmas. But after our examination, the speeds for most of the slow EHs have already reached an equilibrium state. Their velocities are stable around the local minimum of the ion VDF, with no sustained acceleration/deceleration. Although the process of maintaining a stable speed is related to the slope of the ion VDF, the duration of their acceleration/deceleration is too short to be captured by the MMS satellite. Therefore, it is difficult to find a large number of unstable slow EHs for statistical analysis. **Fortunately, we found another event that represents the whole process of slow EHs from unstable to stable, containing 123 slow EHs, to enhance the reliability of our conclusions.** Next, we introduce the event to the reviewer.

Figure R1 presents MMS measurements over 10 seconds interval on August 4, 2017, when the MMS were located at about 20 Earth radii from the Earth in the plasma sheet. A total of 123 slow EHs can be clearly identified by all four MMS satellites in the order of MMS4-MMS1-MMS3-MMS2. The separation of the four MMS satellites in the parallel direction are 5.7 km (MMS4-MMS1), 6.9 km (MMS1-MMS3) and 5.8 km (MMS3-MMS2), respectively. $V_{1j}, j = 2, 3, 4$, represent the average velocities of the slow EHs measured by MMS1 and the other three MMS satellites. Before 16:54:52, the velocities measured by the different satellite pairs separated from each other. The velocities of

these EHs measured by MMS1 and MMS4 is $|V_{14}| > |V_{min}|$, closer to the ion beam. So, these EHs would decelerate due to the positive gradient of the ion beam, which is confirmed by subsequent observation that $|V_{14}| > |V_{min}| > |V_{13}|$. Since $|V_{13}| < |V_{min}|$, the velocities of EHs are closer to the ion core, so these EHs would accelerate due to the negative gradient. Therefore, $|V_{13}| < |V_{12}| \approx |V_{min}|$, the velocities of the EHs eventually stabilize around the local minimum of the ion VDF. From 16:54:48 to 16:54:52, the velocities measured by different satellite pairs gradually converge to the V_{min} , indicating that the velocities of EHs are gradually stable, possibly because the ion beam density is gradually increasing. After 16:54:52, the velocities of EHs stabilize around the V_{min} in the presence of a strong ion beam. This figure shows the whole process of EHs from unstable to stable state, including deceleration at positive gradients and acceleration at negative gradients, which well supports our conclusion and is an important supporting material for our manuscript. For the issues raised by the reviewer, we have tried our best to give corresponding solutions according to the actual situation. We have added a supplementary information and cited it on line 132 in the revised manuscript. We hope our interpretations and modifications can meet the reviewer's requirements.

Figure R1. Whole process of slow EHs from unstable to stable state. **a**, Parallel electric field measured

by four MMS spacecraft. **b**, Ion 1D VDF, integrated from the ion 3D distribution. The black line represents the local minimum of the ion VDF. The different colored dots indicate the EH velocities measured by the different satellite pairs: cyan (MMS1-MMS4), magenta (MMS1-MMS2) and blue (MMS1-MMS3).

On line 137 in the revised manuscript.

“of the ion VDF (see Supplementary Fig. 1 for details),...”

Reviewer #1 -- Major Comment 2:

Line 140: The Authors compare only the distributions of ions with velocities of 1,500-2,000 km/s streaming from region F to region E and claim to identify the velocity variation by a few tens of km/s consistent with theory predictions. However, the distribution functions in regions F and E differ by more than a few tens of km/s at other velocities and this difference does not seem to be consistent with theory. In particular, there are ions propagating from region E to region F, and they should be decelerated according to theory. Do the Authors observe the corresponding effect? It seems that this effect is not observed and, therefore, the Authors' computations do not entirely support the observations. I strongly recommend to do the mapping of the ion distributions from F to E and vice versa and carry out a more careful comparison between theory and observations. Otherwise the comparison between theory and observations is far from being complete.

Response from Authors:

We thank the reviewer for raising this issue, which indeed was not sufficiently addressed in the original manuscript.

According to the reviewer's advice, we made a more nuanced interpretation and conclusion in the revised manuscript. We added the theory prediction curve (dotted red line in Fig. R2) and denoted the reflected ion region (blue box in Fig. R2) in Fig. 4c. The theory prediction is the mapping of the ion distributions (for passing ions, from -1500 km/s to -2000 km/s) from F to E, which is in agreement with that detected by the MMS satellites and further supports our conclusion. Note that there are significantly more reflected ions in region F than those in region E (blue box in Fig. R2). After carefully checking and analysis, the reason is found that the ions in region F are reflected by the positive potential of the EHs (C and D) on both sides, thus causing ion accumulation. The speed of ion accumulation ranges from -760 km/s to -1300 km/s, marked by a blue box in Fig. R2 (Fig. 4c

in the revised manuscript).

As the reviewer said, theoretically the ions from region E to F would indeed be decelerated. However, the velocity of these passing ions is less than 500 km/s (from -500 km/s to 0) in the spacecraft reference. The observation of the low-velocity ion distribution is complicated, so it is difficult to compare with theoretical predictions. For ions moving faster relative to these EHs (>1500 km/s in the spacecraft, relative velocity > 2500 km/s), this deceleration effect is too weak to be observed.

So, the comparison between theory prediction and observation is only for the “passing” region (velocity ranging from -1500 km/s to -2000 km/s in the spacecraft reference and from F to E). To avoid misunderstandings and strengthen our conclusions, we have modified the Fig. 4c and added some explanations on lines 169-174 in the revised manuscript. We hope our interpretations and modifications can meet the reviewer's requirements.

“The theory prediction curve in Fig. 4c is the mapping of the ion distributions (for passing ions, from -1500 km s^{-1} to -2000 km s^{-1}) from region F to region E, which is in agreement with that detected by the MMS satellites and further supports our conclusion. Note that there are significantly more reflected ions in region F than those in region E (blue box in Fig. 4c), because that the ions in region F are reflected by the positive potential of EHs (C and D) on both sides, thus causing ion accumulation.”

Figure R2. 1-D ion velocity distribution function in regions E and F. The red dotted line represents theoretical prediction. The VDF of passing ions and reflected ions are in the black box and blue box, respectively. Regions E and F are the yellow shaded areas in Fig.2a.

Reviewer #1 -- Major Comment 3:

Line 70-71 and 79-82: Using four spacecraft observations the Authors find that the electron hole propagates at 175 deg to local magnetic field at the speed of about 996 km/s. Later in lines 79-82 the Authors assume that electron holes actually propagate strictly parallel to local magnetic field and obtain other speed estimates and infer acceleration/deceleration rates. However what if the electron hole indeed propagates at 175 deg to local magnetic field? In this case the two-spacecraft interferometry applied to whatever pair of MMS spacecraft would provide the speed of 996 km/s and no acceleration/deceleration would be obtained. Thus, there is a fundamental inconsistency in the Authors' analysis and their estimates of electron hole acceleration/deceleration rates are strongly dependent on the electron hole propagation direction. The deviation by only 5 deg off the strictly parallel propagation makes acceleration/deceleration rates equal zero!

Response from Authors:

We thank the reviewer for raising this issue, which indeed was not sufficiently addressed in the

original manuscript.

We are sorry for the misunderstanding caused by the unclear explanation of multispacecraft timing method and two-spacecraft interferometry in the original manuscript. Next, we will answer the reviewer's question by introducing the details of these two methods.

The most important equation of the timing method is as follows:

$$(\mathbf{r}_\alpha - \mathbf{r}_1) \cdot \hat{\mathbf{n}} = V (t_\alpha - t_1)$$

The structure (E-hole in this manuscript) is moving in the direction $\hat{\mathbf{n}}$ with velocity V . During the time $t_\alpha - t_1$ the structure moves along the normal direction a distance $V (t_\alpha - t_1)$ which is equal to the projection of the separation distance $\mathbf{r}_\alpha - \mathbf{r}_1$ onto $\hat{\mathbf{n}}$. Here we take MMS1 as the reference. Due to the tetrahedral structure of the MMS satellites, the four satellites will never be coplanar, so a velocity V and direction $\hat{\mathbf{n}}$ will always be solved. Noted that when we use the timing method, we have assumed that the structure has a fixed velocity and direction through the four satellites MMS1-MMS4. Therefore, it is inevitable that the reviewer will get zero acceleration/deceleration rate along 175 deg to local magnetic field (obtained by timing method), which is a mathematical result.

The timing method has certain limitations, unable to obtain the acceleration/deceleration rates of the structure. The two-spacecraft interferometry is more direct, and multiple speeds can be obtained to estimate the acceleration/deceleration rates, but cannot get the velocity direction.

In the manuscript, we first use the timing method to get the approximate speed and direction (main concern is whether parallel or anti-parallel to local magnetic field), then we calculate the acceleration/deceleration in one-dimensional (parallel/anti-parallel) using two-spacecraft interferometry. Because the focus of our attention and discussion is the interaction between E-holes with ions in the direction of local magnetic field (parallel/anti-parallel).

Once again, we apologize for the misunderstanding caused by the lack of explanation. We have revised and supplemented the original manuscript on lines 95-98 and 211-232 to avoid similar misunderstandings for future readers. We hope our answers and modifications can meet the reviewer's requirements.

“The timing method has certain limitations, unable to obtain the acceleration/deceleration rates of the structure. The two-spacecraft interferometry is more direct, and multiple speeds can be obtained to estimate the acceleration/deceleration rates, but cannot get the velocity direction.”

Reviewer #1 -- Major Comment 4:

Line 75: “the local ion thermal velocity” – Did the Authors compute the thermal speed for the entire ion distribution function? If yes, the distribution is highly non-Maxwellian and, thus, the concept of temperature is inapplicable. The thermal speed of the ion core population would be something reasonable to use in this case.

Response from Authors:

We thank the reviewer for raising this issue, which indeed was not sufficiently addressed in the original manuscript. The ion thermal speed is computed for the entire ion distribution function in the original manuscript. According to the reviewer’s advice, we recalculate and modified to the ion core population thermal speed ($v_{thi} \approx 490$ km/s) on line 87 in the revised manuscript.

Reviewer #1 -- Minor Comments:

--- Line 18: “with acceleration” – Should it be “acceleration/deceleration”? Some of the electron holes in Figure 2 are decelerating.

--- Line 18: “with acceleration on basis of” is better to reformulate as “whose acceleration is revealed using”

--- Line 21: “detection time delay of the MMS satellites” should be “detection time delays between different MMS satellites”

--- Line 23: “, which are consistent with the theoretical predicted results” should be “in accordance with theoretical predictions”

--- Line 45: “electron holes may accelerate and decelerate due to the kinetic features of the ion VDF” - The correct reference is Ref [43].

--- Line 52: “the slow electron hole” should be “slow electron holes”

--- Line 54: “theory analysis” should be “theory”

--- Line 55: “due to the kinetic features” should be “depending on kinetic features”

--- Line 56: “accelerations of electron holes have” should be “acceleration of electron holes has”

--- Line 60: “the net velocity” should be “a net velocity”

--- Line 78: “by four MMS spacecraft.” should be “at four MMS spacecraft”

--- Line 84: “antiparallelly drifts” should be reformulated.

--- Line 88: “decaying” is not a good term here, a better term would be “increasing”

--- Line 114: “changeable” - A better terms is probably “variable”

--- Line 77: "the antiparallel direction of the magnetic field" - better to say "anti-parallel to the magnetic field"

--- Line 173 "In the frame of ion holes" is likely a misprint.

"to the accelerations" should be "to the acceleration"

We thank the reviewer for pointing out these misprints in the original manuscript, which we have corrected on corresponding lines in the revised manuscript.

--- Line 23: “through the decelerating electron holes” – Now I am confused. Are the electron holes accelerating or decelerating? A sentence above the Authors were calling their electron holes “accelerating”.

We thank the reviewer for raising this issue, which indeed was not sufficiently addressed in the original manuscript. We apologize for the ambiguity. We have changed to **acceleration/deceleration** in the revised manuscript.

--- Line 25: “ions passing through them in the same direction as the acceleration of electron holes? – What about the ions passing in the opposite direction?

We thank the reviewer for raising this issue, which indeed was not sufficiently addressed in the original manuscript.

We apologize for the ambiguity. What we want to say with this sentence is electron holes with non-zero acceleration can cause the velocity of ions passing through them to increase in the direction of the acceleration of the electron holes, which is independent of the direction in which the ions pass (the same/opposite direction). We have revised the original manuscript on **lines 21-23** to avoid similar misunderstandings for future readers.

“Therefore, we show that EH with non-zero acceleration can cause velocities of passing ions to increase in the EH acceleration direction.”

--- Line 64: “the plasma sheet” – More information needed to clarify where exactly the spacecraft were located, what the background plasma conditions were etc.

We thank the reviewer for raising this issue, which indeed was not sufficiently addressed in the

original manuscript. According to the reviewer's advice, we add some necessary background plasma conditions on lines 70-73 in the revised manuscript.

“...when the four MMS spacecraft were located in the plasma sheet at the Geocentric Solar Ecliptic (GSE) Coordinates $(-19, -11, 3)R_E$. The local magnetic field is about $(4.5, 8.6, 2.2)$ nT in GSE and the plasma density detected by MMS is below 0.1 cm^{-3} .”

--- Line 83-84: “can obtain three estimated velocities based on four MMS observations” – One can actually get six different pairs of MMS spacecraft. Thus the Author should add three more estimates to their analysis. Does this change the Authors' results?

--- Line 87: “MMS1-MMS3, MMS1-MMS4, and MMS1-MMS2 are” - There is actually six pairs among four spacecraft. Thus the Authors can actually use three more pairs to see whether the velocity increase/decrease is confirmed by the other pairs of spacecraft.

We thank the reviewer for raising this issue. For brevity, we listed three speeds (V13 V14 and V12) in the original manuscript, which are -1250, -1077, and -982 km/s. The E-hole is observed in the order of MMS1-MMS3-MMS4-MMS2. Here we show all six estimated speeds for the reviewer: V13 \approx -1250, V14 \approx -1077, V12 \approx -982, V34 \approx -984, V32 \approx -897, V42 \approx -754 km/s. Although also supporting our results, it is difficult for readers to compare, for example, V12 and V34. In fact, we mainly use the fitting method to get the accelerations/decelerations of E-holes, as shown in Figure 2. The fitting method uses the space separations and time delays of all four MMS spacecraft, which is equivalent to the six estimated speeds from six different pairs of MMS spacecraft. More importantly, it's more intuitive for readers. To avoid similar questions from future readers, we have added an explanation on lines 105-106 in the revised manuscript. We hope our answers and modifications can meet the reviewer's requirements.

“The remaining three estimated velocities of the electron hole also support this result.”

--- Line 94: “in the inertial frame is U” – What “inertial frame” do the Authors mean here? What is “U”? It has not been defined prior to Line 94.

We thank the reviewer for raising this issue, which indeed was not sufficiently addressed in the original manuscript. According to the reviewer's advice, we put the description of “inertial frame” up front and removed the U on lines 111-114 in the revised manuscript.

“The MMS spacecraft moves at a speed of $<1 \text{ km s}^{-1}$, and thus its speed can be neglected compared to the speed of electron holes. Therefore, we regard the spacecraft frame approximately as the inertial frame.”

--- I cannot see in Figure 2 or anywhere before Line 121, where the analysis was done, anything about the acceleration/deceleration rates.

We thank the reviewer for raising this issue, which indeed was not sufficiently addressed in the original manuscript. According to the reviewer’s advice, we have added some additional notes to the captions in Figure 3 (original Figure 2) in the revised manuscript.

On lines 438-439 in the revised manuscript

“The estimated accelerations/decelerations of the four electron holes A-D are 5.7×10^3 , -1.5×10^3 , 1.3×10^4 and $9.6 \times 10^3 \text{ km s}^{-2}$.”

--- Line 141-143: The text in these lines is difficult to understand, English must be improved.

We thank the reviewer for raising this issue, which indeed was not sufficiently addressed in the original manuscript. According to the reviewer’s advice, we improved the sentence in the original manuscript as follows:

On lines 164-169 in the revised manuscript.

“Based on the fitted accelerations/decelerations of the three holes A-C, according to the theoretical calculation (see Methods section, equation (9) in Transit time and velocity difference of ions passing through electron holes for details), for the passing ions (from -1500 to -2000 km s^{-1}) in Fig. 4c, the velocity difference $\Delta v'$ is tens of km s^{-1} along the local magnetic field (Fig. 4b).”

--- Line 179: "Previous studies have suggested that slow electron holes and fast electron holes originate from different plasma instabilities^{5,7,17,27,47,48,49}. However, the positive ion VDF gradient provided by the ion beam can prevent electron holes from self-accelerating into fast electron holes and maintain their speed near the ion thermal velocity, thus being identified as slow electron holes." - I don't see a statement of these two sentences. Yes, there are solid arguments that slow and fast electron holes are produced by different mechanisms, and that does not contradict the second sentence that slow electron holes can remain slow if the distribution function has a local

minimum and the electron hole velocity resides near that minimum without self-acceleration. What is the point of “However” in the beginning of the second sentence and what is the whole point of these two sentences?

We thank the reviewer for raising this issue, which indeed was not sufficiently addressed in the original manuscript.

According to the reviewer’s advice, to avoid ambiguity, we removed the "However".

We apologize for the incompleteness of the statement in our original manuscript. There is a sentence missing in the original manuscript, which we have added in the revised version as follows:

On lines 203-204 in the revised manuscript.

“Therefore, the ion beam is an important factor to maintain the existence of slow electron holes.”

We hope our modifications can meet the reviewer's requirements.

GENERAL COMMENTS FROM REVIEWER #2:

The authors present observations of slow electron phase space holes (EHs) from MMS which include the presence of a passing ion beam. Using multi-spacecraft timing analysis, the EHs are found to be accelerating rather than moving at constant speed. A symmetric EH potential with changing speed has been theorized as capable of accelerating the passing ion population, which appears to be consistent with changes in the ion phase-space density distribution. This is a unique finding which has been investigated thoroughly by the authors and will be useful to others in the field. The methodology is sound and detailed. However, organizational problems and a few notes of clarification hold the paper back from being ready for publication. Most of these may be possible to address in the editing process after minor revisions.

The manuscript – primarily the introduction – is fraught with grammatical errors. None of these directly impact the scientific content, but a corrections pass would be highly beneficial.

Response from Authors:

We thank the reviewer for the positive and encouraging comments, especially that “This is a unique finding which has been investigated thoroughly by the authors and will be useful to others in the field. The methodology is sound and detailed.”. We also thank the reviewer for raising some organizational problems and other comments.

In the following, we address the specific comments point-by-point.

SPECIFIC COMMENTS FROM REVIEWER #2:

Reviewer #2 -- Comment 1:

Organization:

Some of the organizational issues come from Nature Communications’ “methods section last” format, not the author’s design. Even so, references to the methods section are not clear, and make it difficult to parse what was actually done. References via equation numbers or other description at first mention of each method/variable in the text are needed.

Response from Authors:

1. Multispacecraft timing method (e.g. line 70, 134). This method is straightforward, but requires a citation or more thorough description. For example, it is not explained anywhere where the

propagation angle comes from.

We thank the reviewer for raising this issue, which indeed was not sufficiently addressed in the original manuscript. According to the reviewer's advice, we added the description of the multispacecraft timing method to the **Methods section on lines 211-232** in the revised manuscript.

2. *“Theoretical calculation” (e.g. line 143). This is well-explained in the methods section, but at this stage is unclear.*

We thank the reviewer for raising this issue, which indeed was not sufficiently addressed in the original manuscript. According to the reviewer's advice, we added a description of “Theoretical calculation” on **lines 166-167** in the revised manuscript.

“...see Methods section, equation (9) in Transit time and velocity difference of ions passing through electron holes for details...”

3. *EH parameters L , $Udot$, ϕ_0 are used in line 122, but have not yet been defined.*

We thank the reviewer for raising this issue, which indeed was not sufficiently addressed in the original manuscript. According to the reviewer's advice, we added the definition of these EH parameters on **lines 141-142** in the revised manuscript.

“...($L_{||}$, \dot{U} and ϕ_0 , which are the parallel scale, acceleration/deceleration rate and central potential of an electron hole, respectively)...”

4. *Likewise, primed and unprimed velocities are not defined outside of the methods section. For both points 3 and 4, a mention of “see Methods...” is used, but it is still unclear what the reader should look for.*

We thank the reviewer for raising this issue, which indeed was not sufficiently addressed in the original manuscript. According to the reviewer's advice, we added a more detailed description on **lines 83, 145 and 166** in the revised manuscript.

The text of the discussion section serves as more of an introduction/background, and would be much more useful before describing the results. Figure 4 is excellent.

We thank the reviewer for raising this issue, which indeed was not sufficiently addressed in the

original manuscript. According to the reviewer's advice, we moved some text of the discussion section and Figure 4 in the original manuscript before the results section as the overall introduction and background of this research on lines 56-67 in the revised manuscript.

Reviewer #2 -- Comment 2:

Lines 147-148 and conclusions:

It is difficult to say whether deceleration of ions has really been observed. The distribution could just as easily have decreased in density overall. Indeed, looking at the peak of the distribution, it has actually shifted left a bit, suggesting acceleration (if not for the complication of it being outside the 'passing' region, and likely including reflected ion populations). A more nuanced interpretation and conclusion is needed here.

Response from Authors:

We thank the reviewer for raising this issue, which indeed was not sufficiently addressed in the original manuscript.

According to the reviewer's advice, we made a more nuanced interpretation and conclusion in the revised manuscript. We added the theory prediction curve (dotted red line in Fig. R3) and reflected ions region (blue box in Fig. R3) in Fig. 4c. The theory prediction is the mapping of the ion distributions (for passing ions, from -1500 km/s to -2000 km/s) from F to E, which is in agreement with that detected by the MMS satellites and further supports our conclusion. Note that there are significantly more reflected ions in region F than that in region E (blue box in Fig. R3). After careful checking and analysis, the reason is found that the ions in region F are reflected by the positive potential of the EHs (C and D) on both sides, thus causing ion accumulation. The speed of ion accumulation ranges from -760 km/s to -1300 km/s, marked by a blue box in Fig. R3 (Fig. 4c in the revised manuscript). Considering ions outside the "passing" region, it would be too complicated to make a theory prediction. So, the comparison between theory prediction and observation is only for the "passing" region (velocity ranging from -1500 km/s to -2000 km/s and from F to E). To avoid misunderstandings and strengthen our conclusions, we have modified the Fig. 4c and added some explanations on lines 169-174 in the revised manuscript. We hope our interpretations and modifications can meet the reviewer's requirements.

"The theory prediction curve in Fig. 4c is the mapping of the ion distributions (for passing ions,

from -1500 km s^{-1} to -2000 km s^{-1}) from region F to region E, which is in agreement with that detected by the MMS satellites and further supports our conclusion. Note that there are significantly more reflected ions in region F than that in region E (blue box in Fig. 4c), because that the ions in region F are reflected by the positive potential of EHs (C and D) on both sides, thus causing ion accumulation.”

Figure R3. 1-D ion velocity distribution function in regions E and F. The red dotted line represents theoretical prediction. The VDF of passing ions and reflected ions are in the black box and blue box, respectively. Regions E and F are the yellow shaded areas in Fig.2a.

Reviewer #2 -- Comment 3:

Lines 137-139: *Is there a perpendicular component to the E-field? The EHs seem quasi-1D in this instance.*

Response from Authors:

We thank the reviewer for raising this issue, which indeed was not sufficiently addressed in the original manuscript.

There is no perpendicular component to the E-field of the electron holes in this event, which is related to the position of the satellite when it passes through the electron hole. Figure R4 show the

possible waveforms of E_{\parallel} and E_{\perp} for all different locations of the spacecraft. The electron hole is a three-dimensional structure, although satellites sometimes only observe parallel electric field components. In our manuscript, we only consider the properties and effects of electron holes in the parallel direction. To avoid ambiguity, according to the reviewer's advice, we have added some explanations on lines 75-77 and cite the reference in the revised version.

“In this event, the spacecraft did not detect the perpendicular component to the electric field of electron holes, which indicates that the spacecraft's position is close to the center of electron holes⁴⁴.”

Figure R4. Schematic illustration of a two-dimensional solitary potential structure and possible waveforms of E_{\parallel} and E_{\perp} for all different locations of the spacecraft and directions of the ESW [Omura et al., 1999].

Reference:

Omura, Y., H. Kojima, N. Miki, and H. Matsumoto, Two-dimensional electrostatic solitary waves observed by Geotail in the magnetotail, *Adv. Space Res.*, 24, 55-58, 1999.

Reviewer #2 -- Comment 4:

By eye, the EHs in this event appear to be nearly symmetric, but the formula used for ion acceleration assumes perfect symmetry. Is there a slight asymmetry in the data? And how would ion acceleration by that potential compare to the theorized ion acceleration due to the EH motion?

Response from Authors:

We thank the reviewer for raising this issue, which indeed was not sufficiently addressed in the original manuscript.

Parallel E-field waveforms detected by satellites may not be strictly symmetrical, which may be related to the electron distribution inside the EH and the passing trajectory of the satellite. The velocity of passing ions exceeds 1500 km/s, and increasing their speed by 10 km/s requires a net potential difference of more than 150 V. The central potential of the EH in this case is around 300 V, and the net potential difference caused by a slight asymmetry is only a few volts. Therefore, after our careful checking, the passing ion acceleration caused by that potential can be ignored compared to the theorized ion acceleration due to the EH motion. According to the reviewer's advice, to avoid similar confusion for readers, we have added an explanation on lines 174-180 in the revised version.

“The parallel E-field waveforms detected by satellites may not be strictly symmetrical, which is related to the electron distribution inside the EH and the passing trajectory of the satellite. After our careful checking, the net potential difference caused by a slight asymmetry is only a few volts, which has little effect on the velocity of passing ions (velocity range is from -1500 to -2000 km s⁻¹). So, the passing ion acceleration caused by this potential is negligible compared to the theorized ion acceleration due to the EH motion.”

Reviewer #2 -- Comment 5:

In the results section, some indication of the ion/electron Debye length and gyro-radius to compare with the EH scales would be useful.

Response from Authors:

We thank the reviewer for raising this issue, which indeed was not sufficiently addressed in the original manuscript.

According to the reviewer's advice, we have added some indication of the Debye length and ion/electron gyro-radius on lines 158-160 in the revised manuscript to compare with the EH scales.

“The local Debye length is 2.5 km, and the electron and proton gyro-radius are 29 km and 733 km, respectively. So, the parallel scales of electron holes are several Debye lengths and much smaller than the electron and proton gyro-radius.”

GENERAL COMMENTS FROM REVIEWER #3:

This paper uses multi spacecraft MMS observations of symmetric potential electron holes in the magnetotail to verify the theoretical prediction that electron holes with non-zero acceleration can accelerate ions passing through them. Using MMS high resolution data, they investigate the ion distribution functions in these 4 slow electron holes that are decelerating and they show that ions velocities are changing through these holes.

This paper is well written and easy to follow. I will recommend it for publication after authors address my comments.

Response from Authors:

We thank the reviewer for the positive and encouraging comments, especially that “*This paper is well written and easy to follow*”. We also thank the reviewer for highlighting that “*I will recommend it for publication after authors address my comments*”.

In the following, we address the specific comments point-by-point.

SPECIFIC COMMENTS FROM REVIEWER #2:

Reviewer #3 -- Major Comment 1:

Lines 66-68: It is not clear to me how you conclude there is a beam drifting antiparallel to magnetic field from Figure 1b? Could you please explain the figure? Is the beam speed around -1500km/s?

Response from Authors:

We thank the reviewer for raising this issue, which indeed was not sufficiently explained in the original manuscript.

Yes, the ion beam speed is around -1500 km/s. Figure 1b show the 1D ion phase space density distribution function $f(v_{\parallel}, t)$, which is integrated from the 3D ion distribution. The Y-axis is the parallel velocity and the X-axis is time. The 1D ion distribution has a significant enhancement near -1500 km/s after 03:59:03.8 UT and is separated from the ion core (speed below 500 km/s), which is identified as an antiparallel ion beam. According to the reviewer’s advice, we have added explanations in the caption to Figure 2b in the revised manuscript.

“Ion 1D VDF (integrated from the ion 3D distribution, measured at 150 ms cadence by the fast plasma investigation instrument⁴⁶), and the Y-axis is the parallel velocity. The ion 1D VDF has a

significant enhancement near -1500 km s^{-1} after 03:59:03.8 UT and is separated from the ion core (speed below 500 km s^{-1}), which is identified as an antiparallel ion beam.”

Reviewer #3 -- Major Comment 2:

Line 74: So technically they are drifting with the ion beam or close to the ion beam speed?

Response from Authors:

We thank the reviewer for raising this issue, which indeed was not sufficiently explained in the original manuscript.

The drift velocities of these slow electron holes are 800-1000 km/s anti-parallel to the magnetic field, while the ion beam speed is around -1500 km/s , so the electron holes are drifting slightly slower than the ion beam. According to the reviewer’s advice, we have added an explanation on **line 87** in the revised manuscript.

Reviewer #3 -- Minor Comments:

Line 78: consistent with the observed ...

Response from Authors:

We thank the reviewer for pointing out this misprint in the original manuscript, which we have corrected on **line 90** in the revised manuscript.

The Authors have addressed the most of the original comments, but several critical questions about the methodology have not been addressed properly. I reiterate those critical questions again and stress why the previous Authors' responses do not address the raised concerns. Since the methodology is still not entirely justified, the Authors' conclusions are still not supported by the presented analysis. However, the study is indeed very interesting and I would be really glad to see its conclusions fully supported by the methodology.

Major comments

[1] The Authors replied that they cannot estimate the acceleration/deceleration rates from theory. In fact, I provided the incorrect reference to Hutchinson, Phys. Rev. E, 2021, while the correct reference is Hutchinson, 2023, <https://doi.org/10.1063/5.0142790>. Even though the title of this article is "Ion hole equilibrium and dynamics in one dimension" the corresponding analysis is carried out and applicable for both electron and ion holes. Please use Eq. (17) and Eq. (20) to estimate the acceleration/deceleration rates and compare them with the observed rates. Also note that the ion distribution function can be smoothed before computing its gradient, thus there are no problems with obtaining theoretical estimates of the acceleration/deceleration rates using experimental data.

[2] The Authors reply that *"The timing method has certain limitations, unable to obtain the acceleration/deceleration rates of the structure. The two-spacecraft interferometry is more direct, and multiple speeds can be obtained to estimate the acceleration/deceleration rates, but cannot get the velocity direction. In the manuscript, we first use the timing method to get the approximate speed and direction (main concern is whether parallel or anti-parallel to local magnetic field), then we calculate the acceleration/deceleration in one-dimensional (parallel/anti-parallel) using two-spacecraft interferometry. Because the focus of our attention and discussion is the interaction between E-holes with ions in the direction of local magnetic field (parallel/anti-parallel)".* Okay, I understand what the Authors do, but then there is a dilemma – the electron holes either propagate a bit oblique to local magnetic field at a constant speed OR they propagate with acceleration/deceleration strictly anti-parallel to local magnetic field. In other words, if the Authors used slightly oblique propagation in their two-spacecraft interferometry they would clearly obtain ZERO acceleration/deceleration rates. This dilemma need to be resolved, otherwise the Authors conclusions are not supported by the analysis. I guess that the Authors can try solving the overdetermined system of six equations of the multi-spacecraft interferometry to obtain both propagation direction, velocity and acceleration/deceleration rate; what I mean is that Eq. (1) in the manuscript can be actually written for six pairs of MMS spacecraft, rather than only four pairs. The larger number of equations may allow estimating not only the speed, but also the acceleration/deceleration rate.

Minor comments

[1] Line 13-14: *"whether it can finally accelerate ambient electrons (or ions) is quite controversial. Previous theory predicts that net velocity change of passing electrons (or ions) occurs only if the EH have nonzero acceleration"* – This statement is precise only for one-dimensional electron holes. In reality, electron holes

are three-dimensional structures and electron scattering in pitch-angle and energy can occur due to the perpendicular electric field; see, e. g., the following theoretical analysis

<https://ui.adsabs.harvard.edu/abs/2018PhPl...25g2903V/abstract>

[2] Line 77: “*which indicates that the spacecraft's position is close to the center of electron holes*” – That is not necessarily the case and can be simply caused by the fact that the electron holes have pancake shape, which automatically implies that the perpendicular electric field is much smaller than the parallel electric field. In that case, independent of the cut the spacecraft takes through the electron holes, the perpendicular electric field is always smaller than the parallel electric field.

[3] Line 173: “because that the ions” should be “because the ions”

[4] Line 207-208: “at magnetized planets throughout the solar system and beyond.” does not sound relevant. I recommend replacing by “in space plasma.”

Reviewer #2 (Remarks to the Author):

Review of Revision for: NCOMMS-23-26818-T

“Can accelerating slow electron holes exist and cause the net velocity change of ions passing through them?”

Yue Dong, Zhigang Yuan, Shiyong Huang, Zuxiang Xue, Xiongdong Yu, C. J. Pollock, R. B. Torbert, and J. L. Burch

Line numbers here refer to the tracked changes version of the manuscript.

This version of the manuscript is much more clear, and satisfies nearly all of my previous concerns. However, the clarity that came with some of the revisions also raised a few more questions which need to be addressed before this work is ready for publication.

Minor point:

The statement in lines 162-164: “because that the ions in region F are reflected by the positive potential of EHs (C and D) on both sides, thus causing ion accumulation.” is too definitive given no analysis was done to predict the reflected ion population (understandably). It is a plausible conclusion, however, so adding “Plausibly because...” would be sufficient here.

Main Concern:

My comments on the supplementary info below raise a difficulty with this analysis: If the acceleration is $\sim 1e4 \text{ km/s}^2$, and the observation across all spacecraft lasts around 50 ms, then the speed is expected to change by up to $\sim 500 \text{ km/s}$. This is comparable to the gap in the ion VDF between the core and the beam. If the linear acceleration estimate is correct at t_0 , the actual EH acceleration would have to completely change sign by the end of the measurement. How can we be sure that the linear approximation used in fitting is reasonable? And given the EHs will oscillate between the core and beam, are the fit acceleration values reasonable estimates of the ‘true’ average acceleration of the EHs that would ultimately accelerate the passing ions?

The argument in the supplementary discussion is a bit unclear. Is the argument that the difference in velocity from the 3 different baselines for each EH comes from the EH oscillating in velocity over time? In this case, it would make more sense to plot V_{41} , V_{13} , V_{32} which would correspond directly to velocity change over time rather than 14, 12, 13, which all rely on one vertex at MMS1.

One thing is unintuitive about this: why are the velocities always in the same order? If the holes are oscillating, I wouldn't expect each EH encounter to consistently start at $|V_{14}| > |V_{min}|$ and reach $|V_{min}| > |V_{13}|$ every time, but rather be somewhat random. The pattern instead suggests there may be a systematic issue.

To back this claim up, it may be necessary to calculate the bounce period of the EH between the ion core and beam and show it is comparable to the time delays. This might be possible using the results of (Hutchinson et al. 2023), but I will acknowledge is not trivial and likely would merit its own paper.

My suggestion is to remove the detailed analysis of the early part of the observations and focus on the main point: that the velocities stabilize when the ion beam is strong, and might be oscillating or difficult to measure when the ion beam is weak.

I. H. Hutchinson; Ion hole equilibrium and dynamics in one dimension. *Physics of Plasmas* 1 March 2023; 30 (3): 032107. <https://doi.org/10.1063/5.0142790>

Reviewer #3 (Remarks to the Author):

The authors have clarified my questions. I recommend this paper for publication.

Response Letter to Reviewers

We are grateful for the constructive comments on this manuscript (NCOMMS-23-26818A) from all the reviewers. In the text below, each reviewer comment is quoted in italics and is followed by the corresponding detailed response. We have also carefully revised the manuscript and added the Supplementary Information. These updates are highlighted in red in those files. In the text below, the references to these updates are highlighted in a similar way.

GENERAL COMMENTS FROM REVIEWER #1:

The Authors have addressed the most of the original comments, but several critical questions about the methodology have not been addressed properly. I reiterate those critical questions again and stress why the previous Authors' responses do not address the raised concerns. Since the methodology is still not entirely justified, the Authors' conclusions are still not supported by the presented analysis. However, the study is indeed very interesting and I would be really glad to see its conclusions fully supported by the methodology.

Response from Authors:

We thank the reviewer for the positive and encouraging comments, especially that “*the study is indeed very interesting and I would be really glad to see its conclusions fully supported by the methodology.*”. In the following, we address the specific comments point-by-point.

SPECIFIC COMMENTS FROM REVIEWER #1:

Reviewer #1 -- Major Comment 1:

The Authors replied that they cannot estimate the acceleration/deceleration rates from theory. In fact, I provided the incorrect reference to Hutchinson, Phys. Rev. E, 2021, while the correct reference is Hutchinson, 2023, <https://doi.org/10.1063/5.0142790>. Even though the title of this article is “Ion hole equilibrium and dynamics in one dimension” the corresponding analysis is carried out and applicable for both electron and ion holes. Please use Eq. (17) and Eq. (20) to estimate the acceleration/deceleration rates and compare them with the observed rates. Also note that the ion distribution function can be smoothed before computing its gradient, thus there are no problems with obtaining theoretical estimates of the acceleration/deceleration rates using experimental data.

Response from Authors:

Thank the reviewer for raising this issue. We are sorry for missing the important theoretical reference. According to the reviewer's advice, we use Eq. (20) in reference *Hutchinson, 2023* to estimate the acceleration/deceleration rates and compare them with the observed rates. The following is our theoretical calculation taking EH C as an example.

For electron holes, the attracted species is electrons and repelled species is ions ($e \leftrightarrow a$, and $i \leftrightarrow r$). The parameters are $T_a = 7.1 \text{ keV}$, $T_r = 2.4 \text{ keV}$, $\theta = T_r/T_a = 0.34$, $n = 0.06 \text{ cm}^{-3}$ and $\varphi_c = 230 \text{ eV}$, where T_a and T_r are the temperatures of electrons and repelled species, respectively. n and φ_c denote the electron density and the maximum repelled potential energy of potential EH C. We first normalize the ion VDF so that its integral is 1.

$$f_r(v_h) = \frac{f_i(v_h)}{\int f_i(v_h) dv} \quad (R1)$$

The normalized $f_r(v_h)$ is shown in Fig. R1a, the velocity unit is $\sqrt{T_a/m_i}$. Eq. (20) in reference *Hutchinson, 2023* is used to calculate the acceleration/deceleration rate:

$$\dot{v}_h \simeq 4\psi^2 f_r'(v_h)/M_a \quad (R2)$$

Where ψ is the maximum repelled potential energy in units of T_a . $f_r'(v_h)$ is the derivative of $f_r(v_h)$, as shown in Fig. R1b. M_a is the effective mass of the hole, derived from Eq. (21) in reference *Hutchinson, 2023*:

$$M_a \simeq -\frac{16}{3}\psi\sqrt{1+1/\theta} \quad (R3)$$

Then, we calculate the nondimensionalized acceleration/deceleration rate \dot{v}_h according to Eq. (R2). Finally, we convert the dimensionless \dot{v}_h into units of km s^{-2} , noting that the time unit is $\omega_{pa}^{-1} = \sqrt{m_e \epsilon_0 / ne^2}$ and the velocity unit is $\sqrt{T_a/m_i}$.

$$\dot{V}_h = \dot{v}_h \omega_{pa} \sqrt{T_a/m_i} \quad (R4)$$

Fig. R1c shows the theoretical estimates of the acceleration/deceleration rates using experimental data in units of km s^{-2} . **For EH C, the theoretically estimated acceleration rate is around $2 \times 10^4 \text{ km s}^{-2}$, and the observed rate is $1.3 \times 10^4 \text{ km s}^{-2}$.** Similar results were obtained for several other electron holes. Considering measurement errors and theoretical simplification, it is acceptable to have such errors between observations and theoretical predictions. Therefore, it is reasonable that our conclusions are fully supported by the methodology. We have added the comparison between theoretical predictions and observations as supplementary figure 2 and method to the supplementary

information and cited it on line 446-448. Thank the reviewer for raising this issue to enhance the reliability of our conclusions. We hope our reply can meet the reviewer's requirements.

Figure R1. Theoretical estimates of the acceleration/deceleration rates. **a**, the repelled species (ions) distribution function vs particle velocity. **b**, the derivative of the repelled species (ions) distribution function. **c**, the acceleration/deceleration rates obtained from theoretical predictions. The cyan line indicates the observed velocity of the electron hole C. The velocity unit in f_r and its derivative is $\sqrt{T_a/m_i}$.

Reviewer #1 -- Major Comment 2:

The Authors reply that “The timing method has certain limitations, unable to obtain the acceleration/deceleration rates of the structure. The two-spacecraft interferometry is more direct,

and multiple speeds can be obtained to estimate the acceleration/deceleration rates, but cannot get the velocity direction. In the manuscript, we first use the timing method to get the approximate speed and direction (main concern is whether parallel or anti-parallel to local magnetic field), then we calculate the acceleration/deceleration in one-dimensional (parallel/anti-parallel) using two-spacecraft interferometry. Because the focus of our attention and discussion is the interaction between E-holes with ions in the direction of local magnetic field (parallel/anti-parallel)". Okay, I understand what the Authors do, but then there is a dilemma the electron holes either propagate a bit oblique to local magnetic field at a constant speed OR they propagate with acceleration/deceleration strictly anti-parallel to local magnetic field. In other words, if the Authors used slightly oblique propagation in their two-spacecraft interferometry they would clearly obtain ZERO acceleration/deceleration rates. This dilemma need to be resolved, otherwise the Authors conclusions are not supported by the analysis. I guess that the Authors can try solving the overdetermined system of six equations of the multi-spacecraft interferometry to obtain both propagation direction, velocity and acceleration/deceleration rate; what I mean is that Eq. (1) in the manuscript can be actually written for six pairs of MMS spacecraft, rather than only four pairs. The larger number of equations may allow estimating not only the speed, but also the acceleration/deceleration rate.

Response from Authors:

We thank the reviewer for raising this issue, which indeed was not sufficiently explanation in the original manuscript.

We are sorry that we did not clearly explain this problem in our last reply. What we want to express is that when the electron hole has acceleration, the speed and direction obtained by the timing method are not accurate enough. Using the timing method, we default that the electron hole is moving at a constant speed and direction, and the timing method cannot deal with the situation of variable speed. However, when the electron hole has acceleration, the timing method can still get the result. Along the resulting direction, the electron hole behaves as a uniform velocity, which is caused by the limited number of satellites. Four satellites correspond to the requirements of the timing method. When the number of satellites is greater than four, it will be found that the electron hole is not uniform along the resulting direction.

The method is totally incapable of handling relative time differences determined independently

between each of the six different pairs of spacecraft; these time differences may be subject to experimental errors and therefore, in the mathematical sense, they will be mutually inconsistent [1, page 309]. In fact, the three pairs of MMS spacecraft (MMS α -MMS1, $\alpha = 2,3,4$) for Eq. (1) already contain all the information about the spatial topology of MMS spacecraft. Going up to six equations doesn't add any useful information to obtain both direction, velocity and acceleration/deceleration rate, but rather adds more measurement errors. Because the new equations can be expressed as linear combinations of the original equations, such as $\mathbf{r}_3 - \mathbf{r}_2 = (\mathbf{r}_3 - \mathbf{r}_1) - (\mathbf{r}_2 - \mathbf{r}_1)$ and $t_3 - t_2 = (t_3 - t_1) - (t_2 - t_1)$.

But there is no such systematic error for the two-satellite interference. With observations of four satellites, we can obtain three averaged drift speed during three successive time intervals (MMS1-MMS3, MMS3-MMS4, MMS4-MMS2) to drive the acceleration/deceleration rates. The result is close to the fitting method shown in Fig. 3, so we choose the more intuitive fitting method. **In conclusion, limited by the number of satellites, it is not possible to obtain both propagation direction, velocity and acceleration/deceleration rate simultaneously. In our research, we focus on one-dimensional EHs in parallel direction, whose electric fields and interactions with ions/electrons are dominated by parallel direction, so we calculate the acceleration/deceleration rate in parallel direction.** We have added a corresponding explanation on lines 107-115 in the revised manuscript. In addition, according to the reviewer's first major comment, we use Eq. (20) in reference *Hutchinson, 2023 [2]* to estimate the acceleration/deceleration rates and compare them with the observed rates (supplementary figure 2 and method). We understand the reviewer's concerns, and now our conclusions can be supported by the methodology. We hope that our answers and explanations address the reviewer's concerns on this issue.

“Using the timing method, we default that the electron hole is moving at a constant speed and direction. But if the electron hole has an acceleration/deceleration rate, the speed and direction obtained by the timing method are not accurate enough. Therefore, in the following calculation and analysis, we ignore the small deviation in the direction of the timing method result that may be caused by electron hole's acceleration/deceleration. Assume all electron holes drift antiparallel to the local magnetic field and have a fixed acceleration/deceleration rate over the detected period due to the close spatial separation of the MMS satellites.”

Reference:

- [1] Paschmann, G., & Patrick, W. D. Analysis methods for multi-spacecraft data, ISSI Scientific Rep. Series SR-001, ESA/ISSI, International Space Science Institute, Bern, Switzerland, (1998).
- [2] I. H. Hutchinson; Ion hole equilibrium and dynamics in one dimension. Physics of Plasmas 1 March 2023; 30 (3): 032107. <https://doi.org/10.1063/5.0142790>

Reviewer #1 -- Minor Comments:

[1] Line 13-14: “whether it can finally accelerate ambient electrons (or ions) is quite controversial. Previous theory predicts that net velocity change of passing electrons (or ions) occurs only if the EH have nonzero acceleration” – This statement is precise only for one-dimensional electron holes. In reality, electron holes are three-dimensional structures and electron scattering in pitch-angle and energy can occur due to the perpendicular electric field; see, e. g., the following theoretical analysis <https://ui.adsabs.harvard.edu/abs/2018PhPl...25g2903V/abstract>

Response from Authors:

We thank the reviewer for raising this issue, which indeed was not sufficiently addressed in the original manuscript. According to the reviewer’s advice, we changed the statement on lines 14-15 in the revised manuscript to make the expression more precise.

“Previous theory for one-dimensional EHs predicts that net velocity change of passing electrons (or ions) occurs only if the EHs have non-zero acceleration.”

[2] Line 77: “which indicates that the spacecraft’s position is close to the center of electron holes” – That is not necessarily the case and can be simply caused by the fact that the electron holes have pancake shape, which automatically implies that the perpendicular electric field is much smaller than the parallel electric field. In that case, independent of the cut the spacecraft takes through the electron holes, the perpendicular electric field is always smaller than the parallel electric field.

We thank the reviewer for raising this issue, which indeed was not sufficiently addressed in the original manuscript. According to the reviewer’s advice, we changed the statement on lines 75-77 in the revised manuscript to make the expression more precise.

“the perpendicular electric field of electron holes is much smaller than the parallel electric field, due

to the fact that the electron holes have pancake shape.”

[3] *Line 173: “because that the ions” should be “because the ions”*

We thank the reviewer for raising this issue, which indeed was not sufficiently addressed in the original manuscript. According to the reviewer’s advice, we changed the statement on line 178 in the revised manuscript to make the expression more precise.

“plausibly because the ions...”

[4] *Line 207-208: “at magnetized planets throughout the solar system and beyond.” does not sound relevant. I recommend replacing by “in space plasma.”*

We thank the reviewer for raising this issue, which indeed was not sufficiently addressed in the original manuscript. According to the reviewer’s advice, we changed the statement on line 213 in the revised manuscript to make the expression more precise.

“...in space plasma.”

GENERAL COMMENTS FROM REVIEWER #2:

This version of the manuscript is much more clear, and satisfies nearly all of my previous concerns. However, the clarity that came with some of the revisions also raised a few more questions which need to be addressed before this work is ready for publication.

Response from Authors:

We thank the reviewer for the positive and encouraging comments, especially that “*This version of the manuscript is much more clear, and satisfies nearly all of my previous concerns.*”. In the following, we address the specific comments point-by-point.

SPECIFIC COMMENTS FROM REVIEWER #2:

Reviewer #2 – Comment 1:

The statement in lines 162-164: “because that the ions in region F are reflected by the positive potential of EHs (C and D) on both sides, thus causing ion accumulation.” is too definitive given no analysis was done to predict the reflected ion population (understandably). It is a plausible conclusion, however, so adding “Plausibly because...” would be sufficient here.

Response from Authors:

We thank the reviewer for raising this issue, which indeed was not sufficiently addressed in the original manuscript. According to the reviewer’s advice, we changed the statement on **line 178** in the revised manuscript to make the expression more precise.

“**plausibly because the ions...**”

Reviewer #2 – Comment 2:

My comments on the supplementary info below raise a difficulty with this analysis: If the acceleration is $\sim 1e4$ km/s², and the observation across all spacecraft lasts around 50 ms, then the speed is expected to change by up to ~ 500 km/s. This is comparable to the gap in the ion VDF between the core and the beam. If the linear acceleration estimate is correct at t_0 , the actual EH acceleration would have to completely change sign by the end of the measurement. How can we be sure that the linear approximation used in fitting is reasonable? And given the EHs will oscillate between the core and beam, are the fit acceleration values reasonable estimates of the ‘true’ average acceleration of the EHs that would ultimately accelerate the passing ions?

Response from Authors:

We thank the reviewer for raising this concern, which indeed was not sufficiently addressed in the original manuscript. First, the velocity of EHs described in our manuscript is the **average** velocity across all spacecraft, as shown in Fig. 3 (**We have modified the caption for Fig.3 on lines 443-444 in the revised manuscript**). Therefore, the velocity of the EH is on the same side of the ion VDF gap most of the time, except for EH A. As mentioned in the text of the manuscript, the three estimated velocities of the EH C in Fig. 2c corresponding to MMS1-MMS3, MMS1-MMS4, and MMS1-MMS2 are -1250 , -1077 , and -982 km s^{-1} , respectively, which indicates that the drift speed of this electron hole keeps decreasing, rather than drifting at a constant speed. In fact, as the reviewer said, since acceleration/deceleration of the electron hole is related to the gradient of the local ion VDF at the speed of the electron hole, the acceleration/deceleration rates of the hole over the detected period are variable, rather than a fixed constant. Especially, for the drift velocity near the gap in the ion VDF between the core and the beam, the acceleration/deceleration rates approach zero. Secondly, due to the limited satellites, in order to reduce the measurement errors, assuming a fixed acceleration of the hole over the detected period, we use the linear approximations to fit the average acceleration. In addition, we add theoretical predictions of the EH acceleration to the supplementary information (**supplementary figure 2 and method**). The deviation of the results from theoretical prediction and observational fitting is within the acceptable range. Therefore, it is reasonable that the linear approximation used in fitting can represent the “true” average acceleration rate of the EH. Finally, both the fit acceleration rates of EHs and the acceleration of passing ions are supported by the theory, so the conclusions should be reliable.

Reviewer #2 – Comment 3:

The argument in the supplementary discussion is a bit unclear. Is the argument that the difference in velocity from the 3 different baselines for each EH comes from the EH oscillating in velocity over time? In this case, it would make more sense to plot V41, V13, V32 which would correspond directly to velocity change over time rather than 14, 12, 13, which all rely on one vertex at MMS1.

One thing is unintuitive about this: why are the velocities always in the same order? If the holes are oscillating, I wouldn't expect each EH encounter to consistently start at $|V14| > |Vmin|$ and reach $|Vmin| > |V13|$ every time, but rather be somewhat random. The pattern instead suggests there may

be a systematic issue.

To back this claim up, it may be necessary to calculate the bounce period of the EH between the ion core and beam and show it is comparable to the time delays. This might be possible using the results of (Hutchinson et al. 2023), but I will acknowledge is not trivial and likely would merit its own paper. My suggestion is to remove the detailed analysis of the early part of the observations and focus on the main point: that the velocities stabilize when the ion beam is strong, and might be oscillating or difficult to measure when the ion beam is weak.

I. H. Hutchinson; Ion hole equilibrium and dynamics in one dimension. Physics of Plasmas 1 March 2023; 30 (3): 032107. <https://doi.org/10.1063/5.0142790>

Response from Authors:

Very good suggestions. Thanks a lot. The difference in velocity from the 3 different baselines (V41, V13, and V32) for each EH comes from the EH oscillating in velocity over time between ion core and beam. According to the reviewer's advice, we have replaced V14, V12, and V13 with V41, V13, and V32 in Supplementary Figure 1, which would correspond directly to velocity change over time but have little influence on our conclusion. This supplementary event shows the velocities of EHs oscillate between the ion core and beam and the whole process of EHs from unstable to stable, which well supports our conclusion. As the reviewer said, the discussion of the bounce period may be interesting but is not the focus of this work. So, according to the reviewer's suggestion, we remove the detailed analysis of the early part of this observation and focus on the main point: that the velocities stabilize when the ion beam is strong, and might be oscillating or difficult to measure when the ion beam is weak. Corresponding changes are made on **lines 18-20 and 24-26 of the Supplementary Information.**

We thank the reviewer for great suggestions to improve our manuscript, and also thank the reviewer for providing us with an excellent inspiration for our future work.

GENERAL COMMENTS FROM REVIEWER #3:

The authors have clarified my questions. I recommend this paper for publication.

Response from Authors:

We thank the reviewer for highlighting that “*I recommend this paper for publication.*”.

REVIEWERS' COMMENTS

Reviewer #1 (Remarks to the Author):

The Authors have addressed my comments and I do not have any questions about the presented physics. However, the English of this version of the manuscript is much worse than it was previous time. The quality of the English is unacceptably low, which makes it difficult to understand some of the Authors' statements. Below I provide a list of corrections to improve the manuscript, but this list is definitely incomplete. I strongly recommend to substantially improve the English and revise some statements before publication.

Line 19: ", which is"  and

Line 20: " accelerating/deceleration "  accelerating/decelerating

Line 21: "EH"  EHs

Line 22: "velocities of"  the velocity

Line 30: "In addition, the can also exist"  "In addition, they can exist" or "They can also exist"

Line 31: " closely watched"  thoroughly studied

Line 33: "in the plasma physics"  "plasma physics"

Line 44: " indicate"  indicated

Line 45: " interactions"  the interaction

Line 47: "confirm" "revealed" or "showed"

Line 49: "avoid the self-acceleration" avoid acceleration caused by the interaction with ions

Line 54: "In this work, with observations of MMS spacecraft, we show accelerating electron holes"
In this work we present MMS observations of accelerating electrons holes

Line 57: "the slow electron hole with acceleration/deceleration" the accelerating/decelerating
slow electron holes

Line 72: "Coordinates" coordinates

Line 77: "have pancake shape" have the spatial width in the plane perpendicular to the local
magnetic field much larger than the parallel spatial width.

Line 78: " drifts along the antiparallel direction of the local magnetic field" drifts anti-parallel to
the local magnetic field

Line 79: "of the parallel electric field" in the parallel electric field

Line 81: "multispacecraft" four-spacecraft

Line 85: "quasi"  almost

Line 87 and 88: "thermal velocity" thermal speed

Line 90: " anti-parallel to the magnetic field" almost anti-parallel to the local magnetic field

Line 91: " the observed time order of slow electron holes at four MMS spacecraft." the observed occurrence of the slow electron holes aboard four MMS spacecraft.

Line 93: " on the pair" on each pair

Line 94: "the velocity of the solitary structure in the spacecraft frame based on the two-spacecraft interferometry"the spacecraft frame vlocity of the solitary structure

using two-spacecraft interferometry

Line 95: "The timing method " The four-spacecraft timing method

Line 96: "unable to obtain ... rates" since it does not allow us to estimaterate

Line 97: "multiple speeds can be obtained to estimate the acceleration/deceleration rates" allows estimating the acceleration/deceleration rate

Line 98: "but cannot get the velocity direction"but has to presume either strictly parallel or anti-parallel propagation to the local magnetic field.

Line 100: "estimated velocities ...observations" velocity estimates using six pairs of MMS spacecraft.

Line 101: "Considering that the slow electron hole drifts antiparallel to the local magnetic field" Assuming that the slow electron holes propagate strictly

anti-parallel to the local magnetic field

Line 102: " the hole's drift speed " the drift velocity of the slow electron holes

Line 104: "the drift speed" "drift velocity" / The term "speed" can refers only to positive values! /

Line 105: " The remaining three estimated velocities of the electron hole also support this result."
The other three velocity estimates of this electron holes also
support this conclusion.

Line 107: "Using the timing method, we default"  In the four-spacecraft timing method, we
presume that

Lie 108: "has an acceleration/deceleration rate" is accelerating/decelerating

Line 109: "the timing method " the four-spacecraft timing method

Line 110-111: This sentence is not clear and need to be rewritten.

Line 115: "the acceleration of solitary structures" the acceleration rate of the solitary structures

Line 116: "fitted with" fitted to

Line 119: "of electron holes" of the electron holes

Line 121: ". U_0 is the " , while U_0 is the

Line 122: "shows" showed

Line 124: "fitted accelerations/decelerations of four" fitted acceleration/deceleration rate of the
four

Line 126: "all four"-- all the four

line 127: "accelerations/decelerations for" acceleration/deceleration rates for

Line 128: "exactly in the velocity range with" at the

Line 130: "corresponding to" which is consistent with

Line 130: "of ion VDF" of the ion VDF

Line 132: "of other three" for the other three

Line 134: " the passage of" the passage across

Line 136: "to calculate two accelerations" to obtain two estimates of the acceleration rate

Line 137: "at a peak" at the peak

Line 142: "to be stabilized" "to stabilize"

Line 147-149: "We substitute.... environment," I don't understand this part of the sentence. It has to be revised and made clear.

Line 150: "evidence from observations" the evidence

Line 152: "The transit time" The ion transit time across a single electron hole

Line 162: "are derived" were derived

Line 164 & 166: "gyro-radius" gyroradii

Line 167: "estimated values" the estimated values

Line 167: " would not have just passed" did not necessarily passed

Line 169: "from the ions"- remove this.

Line 170: "accelerations/decelerations" acceleration/deceleration rates

Line 174: " theory prediction curve" the theoretical curve

Line 175: " distributions" distribution

Line 176: " that detected by the" the ion VDF observed aboard

Line 180-181: ", which is related to the electron distribution inside the EH and the passing trajectory of the satellite."

This statement is very confusing. The net potential across EHs may appear because of asymmetric ion reflection, but not because of the electron distribution within EH or because of spacecraft trajectory through EH.

Line 181-182: " After our careful checking, the net" After a careful analysis we found that the net

Line 182: "difference" difference across the electron holes that is

Line 185: "the theorized ion acceleration due to the EH motion." Did you mean "the theoretical acceleration due to the net potential drop"?

Line 193: "the multi-satellite simultaneous detection" simultaneous multi-satellite measurements

Line 202: "parallel to the acceleration are exchanged from those in the antiparallel direction" - I don't understand this statement. Do you mean that the ion propagating parallel and anti-parallel exchange energies? That would be strange.

Line 207: " Positive ion VDF gradient provided by the ... us being identified as slow electron holes. Therefore, the ion beam is an important factor to maintain the existence of slow electron holes" - These are not the results of the Author's study. Both statements have been already proved by Kamaletdinov et al., PRL, 2021.

Reviewer #2 (Remarks to the Author):

The authors have addressed all of my concerns. This paper can be published in its current form.

Response Letter to Reviewers

We are grateful for the constructive comments on this manuscript (NCOMMS-23-26818B) from all the reviewers. In the text below, each reviewer comment is quoted in italics and is followed by the corresponding detailed response. We have also carefully revised the manuscript and added the Supplementary Information. These updates are highlighted in red in those files. In the text below, the references to these updates are highlighted in a similar way.

GENERAL COMMENTS FROM REVIEWER #1:

The Authors have addressed my comments and I do not have any questions about the presented physics. However, the English of this version of the manuscript is much worse than it was previous time. The quality of the English is unacceptably low, which makes it difficult to understand some of the Authors' statements. Below I provide a list of corrections to improve the manuscript, but this list is definitely incomplete. I strongly recommend to substantially improve the English and revise some statements before publication.

Response from Authors:

We thank the reviewer for providing the list of corrections to improve the manuscript. In the following, we address the specific comments point-by-point.

SPECIFIC COMMENTS FROM REVIEWER #1:

Line 19: ", which is"  and

Thanks a lot, we have corrected it in the revised manuscript.

Line 20: "accelerating/deceleration " accelerating/decelerating

Thanks a lot, we have corrected it in the revised manuscript.

Line 21: "EH" EHs

Thanks a lot, we have corrected it in the revised manuscript.

Line 22: "velocities of" the velocity

Thanks a lot, we have corrected it in the revised manuscript.

Line 30: "In addition, the can also exist" "In addition, the can exist" or "They can also exist"

Thanks a lot, we have corrected it in the revised manuscript.

Line 31: "closely watched" thoroughly studied

Thanks a lot, we have corrected it in the revised manuscript.

Line 33: "in the plasma physics" "plasma physics"

Thanks a lot, we have corrected it in the revised manuscript.

Line 44: "indicate" indicated

Thanks a lot, we have corrected it in the revised manuscript.

Line 45: "interactions" the interaction

Thanks a lot, we have corrected it in the revised manuscript.

Line 47: "confirm" "revealed" or "showed"

Thanks a lot, we have corrected it in the revised manuscript.

Lie 49: *"avoid the self-acceleration" avoid acceleration caused by the interaction with ions*

Thanks a lot, we have corrected it in the revised manuscript.

Line 54: *"In this work, with observations of MMS spacecraft, we show accelerating electron holes"-> In this work we present MMS observations of accelerating electrons holes*

Thanks a lot, we have corrected it in the revised manuscript.

Line 57: *"the slow electron hole with acceleration/deceleration" the accelerating/decelerating slow electron holes*

Thanks a lot, we have corrected it in the revised manuscript.

Line 72: *"Coordinates" coordinates*

Thanks a lot, we have corrected it in the revised manuscript.

Line 77: *"have pancake shape" have the spatial width in the plane perpendicular to the local magnetic field much larger than the parallel spatial width.*

Thanks a lot, we have corrected it in the revised manuscript.

Line 78: *" drifts along the antiparallel direction of the local magnetic field" drifts anti-parallel to the local magnetic field*

Thanks a lot, we have corrected it in the revised manuscript.

Line 79: *"of the parallel electric field" in the parallel electric field*

Thanks a lot, we have corrected it in the revised manuscript.

Line 81: *"multispacecraft" four-spacecraft*

Thanks a lot, we have corrected it in the revised manuscript.

Line 85: *"quasi"  almost*

Thanks a lot, we have corrected it in the revised manuscript.

Line 87 and 88: *"thermal velocity" thermal speed*

Thanks a lot, we have corrected it in the revised manuscript.

Line 90: *" anti-parallel to the magnetic field" almost anti-parallel to the local magnetic field*

Thanks a lot, we have corrected it in the revised manuscript.

Line 91: *" the observed time order of slow electron holes at four MMS spacecraft." the observed occurrence of the slow electron holes aboard four MMS spacecraft.*

Thanks a lot, we have corrected it in the revised manuscript.

Line 93: *" on the pair" on each pair*

Thanks a lot, we have corrected it in the revised manuscript.

Lie 94: *"the velocity of the solitary structure in the spacecraft frame based on the two-spacecraft interferometry"the spacecraft frame vlocity of the solitary structure using two-spacecraft interferometry*

Thanks a lot, we have corrected it in the revised manuscript.

Line 95: *"The timing method " The four-spacecraft timing method*

Thanks a lot, we have corrected it in the revised manuscript.

Line 96: *"unable to obtain ... rates" since it does not allow us to estimaterate*

Thanks a lot, we have corrected it in the revised manuscript.

Line 97: *"multiple speeds can be obtained to estimate the acceleration/deceleration rates" allows estimating the acceleration/deceleration rate*

Thanks a lot, we have corrected it in the revised manuscript.

Line 98: *"but cannot get the velocity direction"but has to presume either strictly parallel or anti-parallel propagation to the local magnetic field.*

Thanks a lot, we have corrected it in the revised manuscript.

Line 100: "estimated velocities ...observations" velocity estimates using six pairs of MMS spacecraft.

Thanks a lot, we have corrected it in the revised manuscript.

Line 101: "Considering that the slow electron hole drifts antiparallel to the local magnetic field"-> Assuming that the slow electron holes propagate strictly anti-parallel to the local magnetic field

Thanks a lot, we have corrected it in the revised manuscript.

Line 102: " the hole's drift speed " the drift velocity of the slow electron holes

Thanks a lot, we have corrected it in the revised manuscript.

Line 104: "the drift speed" "drift velocity" / The term "speed" can refer only to positive values!
/

Thanks a lot, we have corrected it in the revised manuscript.

Line 105: " The remaining three estimated velocities of the electron hole also support this result."-> The other three velocity estimates of this electron holes also support this conclusion.

Thanks a lot, we have corrected it in the revised manuscript.

Line 107: "Using the timing method, we default"  In the four-spacecraft timing method, we presume that

Thanks a lot, we have corrected it in the revised manuscript.

Line 108: "has an acceleration/deceleration rate" is accelerating/decelerating

Thanks a lot, we have corrected it in the revised manuscript.

Line 109: "the timing method " the four-spacecraft timing method

Thanks a lot, we have corrected it in the revised manuscript.

Line 110-111: This sentence is not clear and need to be rewritten.

Thanks a lot, we have rewritten this sentence in the revised manuscript.

Line 115: "the acceleration of solitary structures" the acceleration rate of the solitary structures

Thanks a lot, we have corrected it in the revised manuscript.

Line 116: "fitted with" fitted to

Thanks a lot, we have corrected it in the revised manuscript.

Line 119: "of electron holes" of the electron holes

Thanks a lot, we have corrected it in the revised manuscript.

Line 121: ". U_0 is the " , while U_0 is the

Thanks a lot, we have corrected it in the revised manuscript.

Line 122: "shows" showed

Thanks a lot, we have corrected it in the revised manuscript.

Line 124: "fitted accelerations/decelerations of four" fitted acceleration/deceleration rate of the four

Thanks a lot, we have corrected it in the revised manuscript.

Line 126: "all four"-- all the four

Thanks a lot, we have corrected it in the revised manuscript.

line 127: "accelerations/decelerations for" acceleration/deceleration rates for

Thanks a lot, we have corrected it in the revised manuscript.

Line 128: "exactly in the velocity range with" at the

Thanks a lot, we have corrected it in the revised manuscript.

Line 130: "corresponding to" which is consistent with

Thanks a lot, we have corrected it in the revised manuscript.

Line 130: "of ion VDF" of the ion VDF

Thanks a lot, we have corrected it in the revised manuscript.

Line 132: "of other three" for the other three

Thanks a lot, we have corrected it in the revised manuscript.

Line 134: "the passage of" the passage across

Thanks a lot, we have corrected it in the revised manuscript.

Line 136: "to calculate two accelerations" to obtain two estimates of the acceleration rate

Thanks a lot, we have corrected it in the revised manuscript.

Line 137: "at a peak" at the peak

Thanks a lot, we have corrected it in the revised manuscript.

Line 142: "to be stabilized" "to stabilize"

Thanks a lot, we have corrected it in the revised manuscript.

Line 147-149: "We substitute.... environment," I don't understand this part of the sentence. It has to be revised and made clear.

Thanks a lot. On line 148-150, we have rewritten this sentence in the revised manuscript as "We perform theoretical calculations based on actual plasma parameters ($L_{||}$, \dot{U} and Φ , which are the parallel scale, acceleration/deceleration rate and central potential of an electron hole, respectively)"

Line 150: "evidence from observations" the evidence

Thanks a lot, we have corrected it in the revised manuscript.

Line 152: "The transit time" The ion transit time across a single electron hole

Thanks a lot, we have corrected it in the revised manuscript.

Line 162: "are derived" were derived

Thanks a lot, we have corrected it in the revised manuscript.

Line 164 & 166: "gyro-radius" gyroradii

Thanks a lot, we have corrected it in the revised manuscript.

Line 167: "estimated values" the estimated values

Thanks a lot, we have corrected it in the revised manuscript.

Line 167: "would not have just passed" did not necessarily passed

Thanks a lot, we have corrected it in the revised manuscript.

Line 169: "from the ions"- remove this.

Thanks a lot, we have corrected it in the revised manuscript.

Line 170: "accelerations/decelerations" acceleration/deceleration rates

Thanks a lot, we have corrected it in the revised manuscript.

Line 174: "theory prediction curve" the theoretical curve

Thanks a lot, we have corrected it in the revised manuscript.

Line 175: "distributions" distribution

Thanks a lot, we have corrected it in the revised manuscript.

Line 176: "that detected by the" the ion VDF observed aboard

Thanks a lot, we have corrected it in the revised manuscript.

Line 180-181: ", which is related to the electron distribution inside the EH and the passing trajectory of the satellite."

This statement is very confusing. The net potential across EHs may appear because of asymmetric ion reflection, but not because of the electron distribution within EH or because of spacecraft trajectory through EH.

Thanks a lot, we have corrected this statement in the revised manuscript.

Line 181-182: " After our careful checking, the net" After a careful analysis we found that the net

Thanks a lot, we have corrected it in the revised manuscript.

Line 182: "difference" difference across the electron holes that is

Thanks a lot, we have corrected it in the revised manuscript.

Line 185: "the theorized ion acceleration due to the EH motion." Did you mean "the theoretical acceleration due to the net potential drop"?

Thanks a lot, “the theoretical ion acceleration” in this sentence means that the net velocity change of passing ions due to the acceleration/deceleration rate of the electron holes. To avoid ambiguity, we have changed this sentence in the revised manuscript.

Line 193: "the multi-satellite simultaneous detection" simultaneous multi-satellite measurements

Thanks a lot, we have corrected it in the revised manuscript.

Line 202: "parallel to the acceleration are exchanged from those in the antiparallel direction" - I don't understand this statement. Do you mean that the ion propagating parallel and anti-parallel exchange energies? That would be strange.

Thanks a lot. “Parallel and anti-parallel” means parallel (or anti-parallel) to the acceleration direction of the electron holes rather than parallel (anti-parallel) to the local magnetic field. In addition, the velocity is relative velocity in the hole frame. So, parallel (or anti-parallel) means that the relative velocity is parallel (or anti-parallel) to the acceleration direction of the electron holes. In other words, ions with velocity faster or slower than the electron holes. We have added the reference in the revised manuscript.

Line 207: " Positive ion VDF gradient provided by the ... us being identified as slow electron holes. Therefore, the ion beam is an important factor to maintain the existence of slow electron holes" - These are not the results of the Author's study. Both statements have been already proved by Kamaletdinov et al., PRL, 2021.

Thanks a lot, we have added the reference in the revised manuscript.

GENERAL COMMENTS FROM REVIEWER #2:

The authors have addressed all of my concerns. This paper can be published in its current form.

Thanks for the reviewer's encouragement and valuable comments, which greatly improved our paper.